# Conformalized Predictions in Hypergraph Neural Networks via Contrastive Learning

## Abstract

Hypergraph representation learning has gained immense popularity over the last few years due to its applications in real-world domains like social network analysis, recommendation systems, biological network modeling, and knowledge graphs. However, hypergraph neural networks (HGNNs) lack rigorous uncertainty estimates, which limits their deployment in critical applications where the reliability of predictions is crucial. To bridge this gap, we propose Contrastive Conformal HGNN (CCF-HGNN) that jointly accounts for aleatoric and epistemic uncertainties in hypergraph-based models for guaranteed and robust uncertainty estimates. CCF-HGNN accounts for epistemic uncertainty in HGNN predictions by producing a prediction set that leverages the topological structure and provably contains the true label with a pre-defined coverage probability. It also accounts for aleatoric uncertainty by leveraging contrastive learning on the structure of the hypergraph. To enhance the power of the predictions, CCF-HGNN performs an additional auxiliary task of hyperedge degree prediction with an end-to-end differentiable sampling-based approach. Extensive experiments on real-world hypergraph datasets demonstrate the superiority of CCF-HGNN by improving the efficiency of prediction sets while maintaining valid coverage.

## 1 Introduction

Network-structured data underpins a broad spectrum of scientific and real-world applications, ranging from social interactions Newman (2003) and recommender systems Ying et al. (2018) to biological networks Zhang et al. (2022) and knowledge graphs Nickel et al. (2015). This has fueled the rapid growth of graph-based machine learning, where graph neural networks (GNNs) have emerged as a dominant paradigm for learning from relational data Kipf & Welling (2016); Hamilton et al. (2017); Velickovic et al. (2017). More recently, attention has shifted towards *hypergraph representation learning*, which extends beyond pairwise relations to model higher-order interactions, thereby offering a more faithful abstraction for many complex systems Battaglia et al. (2018); Zhang et al. (2018b). The expressive power of hypergraphs has led to applications across diverse domains, including healthcare (e.g., multiple patients sharing a room) Xu et al. (2022); Choudhuri et al. (2025a); Xu et al. (2023), social networks (e.g., users joining groups or channels) Li et al. (2013), bioinformatics Tian et al. (2009), and cyber-security Lin et al. (2024). To exploit these structures, hypergraph neural networks (HGNNs) have been developed with specialized message-passing and aggregation mechanisms Feng et al. (2019); Yadati et al. (2019); Bai et al. (2021), demonstrating superior performance when group-wise relations, rather than dyadic links, are essential.

The evolution of HGNNs has closely paralleled that of GNNs. Early work, such as HGNN Feng et al. (2019), adapted the message-passing framework of GCN Kipf & Welling (2016), while HCHA Bai et al. (2021) extended the attention mechanism of GAT Velickovic et al. (2017) to hypergraphs. More recent efforts have introduced advanced ideas, including multiset functions Chien et al. (2021), network diffusion Wang et al. (2023a), energy-based formulations Wang et al. (2023b), and implicit modeling Li et al. (2025); Choudhuri et al. (2025b). Despite these innovations, a key limitation persists: *existing HGNNs provide no mechanism to quantify predictive uncertainty*. This omission is particularly problematic in high-stakes domains, where decisions require not only accuracy but also calibrated confidence. A principled solution is to construct *prediction sets* that guarantee high-probability coverage for each sample. While numerous uncertainty quantification methods have been proposed in the broader machine learning literature Guo et al. (2017); Zhang et al. (2020); Hsu et al.

(2022); Zhang et al. (2018a); Gal & Ghahramani (2016); Trivedi et al. (2023); Lakshminarayanan et al. (2017), they generally lack rigorous coverage guarantees—i.e., assurances that the true label lies within the predicted set with the desired probability.

The field of conformal prediction, pioneered by Vovk et al. (2005), provides a principled framework for constructing prediction sets with rigorous, finite-sample coverage guarantees under minimal distributional assumptions. By calibrating nonconformity scores on held-out data, conformal prediction methods ensure that the true label is included in the prediction set with a user-specified probability (e.g., $1 - \alpha$), regardless of the underlying data distribution. This property has fueled widespread adoption in areas such as computer vision Angelopoulos & Bates (2021), natural language processing Kumar et al. (2023), and time-series forecasting Stankeviciute et al. (2021); Zaffran et al. (2022); Gibbs & Candes (2021). Conformal prediction has gained immense popularity in graph representation learning, with frameworks aimed at quantifying uncertainty in inductive Clarkson (2023); Zargarbashi & Bojchevski (2023) and transductive Zargarbashi et al. (2023); Huang et al. (2024) node classification and edge/link prediction Luo & Colombo (2024); Zhao et al. (2024); Choudhuri et al. (2025c).

While conformal prediction provides coverage guarantees, it explicitly quantifies epistemic uncertainty (i.e., model-specific uncertainty) while ignoring aleatoric uncertainty (i.e., data uncertainty). Moreover, as highlighted in Table 1, uncertainty quantification in hypergraph representation learning has not been studied before. Quantifying both sources of uncertainty is particularly important in hypergraphs, as data often appears in the form of higher-order relationships (e.g., co-authorship, biochemical complexes, and co-purchases) that are prone to noise, sparsity, and lack of standardization, limiting effective model training in practice. Unlike graphs, where uncertainty typically stems from missing or spurious edges, hypergraphs are subject to structural ambiguity in the semantics of multi-way relations (e.g., "all authors of a paper" or "all participants in a discussion"), which introduces additional sources of noise. Furthermore, in node classification tasks, a single mislabeled node does not merely affect its immediate neighbors, as in graphs, but can propagate errors to every node within the same hyperedge, significantly amplifying the impact of label noise.

Table 1: Comparison of the features of the prior works. Our proposed framework jointly accounts for epistemic and aleatoric uncertainties while maintaining valid marginal coverage.

| Method | Coverage | Epistemic | Aleatoric |
|---|---|---|---|
| TS and VS Guo et al. (2017) | ✗ | ✗ | ✓ |
| ETS Zhang et al. (2020) | ✗ | ✗ | ✓ |
| CF-GNN Huang et al. (2024) | ✓ | ✓ | ✗ |
| Ours | ✓ | ✓ | ✓ |

To address these challenges, we introduce *Contrastive Conformal Hypergraph Neural Network* (**CCF-HGNN**), an end-to-end framework that jointly models aleatoric and epistemic uncertainty in hypergraph representation learning. Our contributions are as follows:

- To the best of our knowledge, this is the first work that combines aleatoric uncertainty (contrastive augmentation- aided learning) and epistemic uncertainty (conformal prediction).

- We propose an auxiliary hyperedge-degree prediction task to our overall conformal training algorithm to boost the power of the hypergraph representations. We additionally propose an efficient computational method to sample the important hyperedges based on the augmentation strategy and perform the hyperedge-degree prediction task only on those hyperedges.

- We provide theoretical evidence that guarantees that the joint modeling of epistemic and aleatoric uncertainties is both efficient (ie, the predictive bands returned are shorter) and effective (empirical coverage provably exceeds the given confidence level).

- Extensive experiments on several real-world hypergraph datasets for uncertainty quantification in the node classification task demonstrate the overall utility of our method.

## 2 PRELIMINARIES

Let $H = (\mathcal{V}, \mathcal{E}, \mathcal{X}, \mathcal{Y})$ be a hypergraph, where $\mathcal{V}$ is a set of nodes, $\mathcal{E}$ is a set of hyperedges, and $\mathcal{X} = \{\mathbf{x}_v\}_{v \in \mathcal{V}}$ is the set of node attributes, where $\mathbf{x}_v \in \mathbb{R}^d$ is a $d$-dimensional feature vector for node $v \in \mathcal{V}$. Let $\mathcal{Y} = \{y_v\}_{(v) \in \mathcal{V}}$ be the set of node labels. Our paper focuses on classification problems,

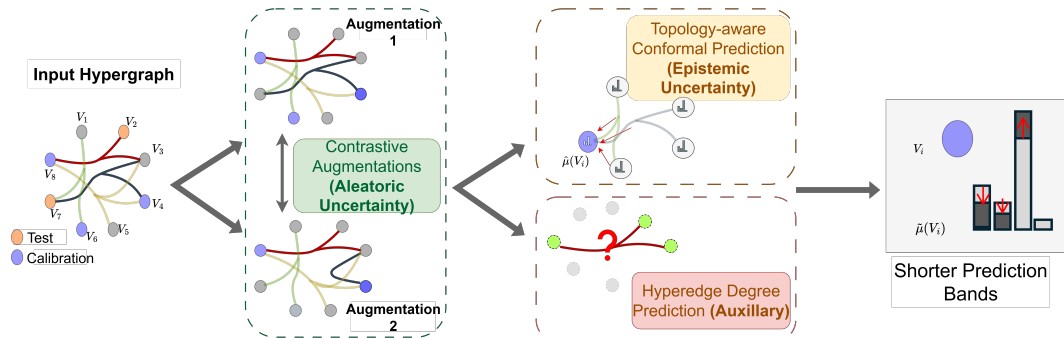

Figure 1: **Contrastive Conformal Hypergraph Neural Network:** The overall framework minimizes three losses: 1) Contrastive Loss: Structural alterations generate multiple views of the hypergraph, encouraging the model to learn invariant representations. 2) Conformal Inefficiency Loss: Topology-aware conformal loss ensures similarity in uncertainties of a node based on its local neighbors (nodes that share hyperedges). 3) Degree Loss: Predicting the hyperedge degree of a sample of hyperedges to guide the model to learn the structure. This leads to shorter and more confident prediction bands.

but our theory and method naturally extend to regression problems. To perform point predictions, we are given a mean estimator $\hat{\mu}$ that predicts the node label $\hat{y}_v$ given the node embedding $x_v$.

## 2.1 TRANSDUCTIVE SETTING

We focus on the transductive node classification problem with a random data split akin to Huang et al. (2024). In this setting, we partition the node labels into three disjoint sets: $\mathcal{Y}_{\text{train}}$, $\mathcal{Y}_{\text{cal}}$, and $\mathcal{Y}_{\text{test}}$. This leads to *training* data $D_{\text{train}} = (\mathcal{V}, \mathcal{E}, \mathcal{X}, \mathcal{Y}_{\text{train}})$, *calibration* data $D_{\text{cal}} = (\mathcal{V}, \mathcal{E}, \mathcal{X}, \mathcal{Y}_{\text{cal}})$, and *testing* data $D_{\text{test}} = (\mathcal{V}, \mathcal{E}, \mathcal{X}, \mathcal{Y}_{\text{test}})$. In particular, during training, the model can access $\mathcal{V}, \mathcal{E}, \mathcal{X}$, but only the training labels $\mathcal{Y}_{\text{train}}$ are revealed to the model. Abusing the notation, we use $\mathcal{V}_{\text{train}}$ to denote elements of $\mathcal{V}$ for which the node labels are in $\mathcal{Y}_{\text{train}}$. We follow the same notation throughout the paper. After training, the calibration data $\{y_v\}_{v \in \mathcal{V}_{\text{cal}}}$ is used to construct uncertainty estimates. Finally, we predict the uncertainty bands for the remaining nodes (i.e., $\mathcal{V}_{\text{test}}$).

## 2.2 MEAN ESTIMATOR: HYPERGRAPH NEURAL NETWORK

Hypergraph Neural Networks (HGNNs) are powerful machine learning models that leverage the high-order network structure during message passing. Unlike traditional graph neural networks that only aggregate pairwise information, HGNNs can handle the complexity of hypergraphs, where relationships between nodes are generalized beyond pairwise connections. Like Graph Neural Networks (GNNs), HGNNs aggregate neighborhood information Bai et al. (2021); Feng et al. (2019) via a sequence of propagation layers where each layer consists of a Message Passing Step, and a Node Update Step. Further details about the propagation steps are provided in the Appendix A.1.

## 2.3 CONFORMAL PREDICTION

In this work, we focus on split conformal prediction Vovk et al. (2005), which proceeds in four primary steps. Given a miscoverage rate $\alpha \in [0, 1]$, the steps are: **(1) Training:** Train the mean estimator $\hat{\mu}$ on the training data $D_{\text{train}}$. **(2) Calibration:** For each node $v$ in $\mathcal{V}_{\text{cal}}$, compute the non-conformity scores (heuristic notion of how off the prediction is from the true label) $\{V(\mathbf{x}_v, y_v)\}_{v \in \mathcal{V}_{\text{cal}}}$ and create an empirical distribution from the scores. **(3) Quantile Computation:** Compute the $(1 - \alpha)^{\text{th}}$ quantile $\hat{Q}_{1-\alpha}$ of the distribution $\frac{1}{|\mathcal{V}_{cal}|+1} \sum_{v \in \mathcal{V}_{cal}} \delta_{V_v} + \delta_{\infty}$, where $\delta_a$ is Dirac Delta distribution at point $a$, and $V_v$ is shorthand for $V(\mathbf{x}_v, y_v)$. **(4) Band Computation:** Given a test node $v$ and corresponding feature $\mathbf{x}_v$, a prediction set/interval $\hat{C}(\mathbf{x}_v) = \{y \in \mathcal{Y} : V(\mathbf{x}_v, y) \leq \hat{Q}_{1-\alpha}\}$ is constructed. The notion of transferring the prediction bands computed on the calibration data to the points in test data relies on the following permutation invariance assumption Huang et al. (2024); Zargarbashi & Bojchevski (2023).

**Assumption 1.** *For any permutation $\pi$ on the calibration and test nodes, the non-conformity score $V$ obeys*

$$V(\mathbf{x}_v, y_v; \{y_a\}_{(a)\in\mathcal{V}_{train\cup cal}}, \mathcal{X}, \mathcal{V}, \mathcal{E}) = V(\mathbf{x}_v, y_v; \{y_a\}_{(a)\in\mathcal{V}_{train\cup cal}}, \mathcal{X}, \mathcal{V}_\pi, \mathcal{E})$$

*This means that the non-conformity scores of nodes in a hypergraph $H$ are exchangeable.*

Assumption 1 imposes the permutation invariance condition for the HGNN training to later compute the non-conformity scores for node prediction, which means that the model output/non-conformity score is invariant to permuting the order of the calibration and test nodes on the hypergraph. HGNNs do not rely on the ordering of the nodes, hence they typically satisfy the assumption.

**Lemma 1.** *(Coverage Guarantee for Conformal Inference) Vovk et al. (2005); Tibshirani et al. (2019) Under Assumption 1, for any $\alpha > 0$, the confidence band returned by the conformal inference algorithm satisfies:*

$$\mathbb{P}(y_v \in \hat{C}_{1-\alpha}(\mathbf{x}_v)) \geq 1 - \alpha \tag{1}$$

*where the probability is taken over the calibration fold $D_{cal}$ and the testing point $(\mathbf{x}_v, y_v)$.*

Here, $\mathbb{P}(y_v \in \hat{C}_{1-\alpha}(\mathbf{x}_v))$ denotes the **coverage**, i.e., the probability that the true label $y_v$ lies in the predictive band.

## 3    OUR METHOD

In this section, we propose our method, Contrastive Conformal Hypergraph Neural Network (CCF-HGNN), which aims to reduce the size of the predictive band length while maintaining coverage for hypergraph neural networks. The main idea is to boost the APS and RAPS scores (see section 4.1) with the help of local topological information and account for data-noise in the form of contrastive augmentations.

### 3.1    COMPUTING DIFFERENTIABLE INEFFICIENCY LOSS

Instead of using pairwise local topological information as done by Huang et al. (2024), our work uses high-order local topological information that goes beyond homophily or other aggregation mechanisms (like mean, sum, etc.). To implement this idea, we use a separate HGNN learner $\tilde{\mu}$ parameterized by the weights $\vartheta$ for the same hypergraph network $H$ with node features initialized by $\hat{\mu}(\mathcal{X})$. Here $\hat{\mu}(\cdot)$ denotes the mean estimator that has been used during the training process. Given $\tilde{\mu}(\mathcal{X}) = \text{HGNN}_\vartheta(\hat{\mu}(\mathcal{X}), H)$, and a target miscoverage rate $\alpha$, we partition the calibration data $D_{\text{cal}}$ into $D_{\text{corr-cal}}$ (correction subset) and $D_{\text{cal-test}}$ (testing subset) compute a differentiable loss in the following steps: 1) **Differentiable Quantile Computation:** Compute the smooth differentiable quantile $\hat{\eta} = \text{DiffQuantile}(\{V(\mathbf{x}_i, y_i) \mid i \in \mathcal{D}_{\text{corr-cal}}\})$ on $D_{\text{corr-cal}}$. 2) **Inefficiency Proxies Computation:** Construct a differentiable proxy of the miscoverage on $D_{\text{cal-test}}$ by using $D_{\text{corr-cal}}$ as calibration data. For class $k$ and node $i$ in $D_{\text{cal-test}}$, the non-conformity score is given as $V(\mathbf{x}_i, k)$ (as per APS and RAPS scores). The inefficiency proxy will thus be $c_i = \sigma\left(\frac{V(\mathbf{x}_i, k) - \hat{\eta}}{\tau_1}\right)$, where $\sigma(\cdot)$ denotes the sigmoid function and $\tau_1$ denotes the temperature hyperparameter Stutz et al. (2022). 3) **Overall Loss Computation:** Compute the overall inefficiency loss as an average of the inefficiency proxies $\mathcal{L}_{\text{Ineff}} = \frac{1}{m} \sum_{i \in \mathcal{D}_{\text{cal-test}}} \frac{1}{|\mathcal{Y}|} \sum_{k \in \mathcal{Y}} c_i$.

The proof that the inefficiency loss is exchangeable simply follows the proof of the same theorem given in Huang et al. (2024) as our setup also operates on the transductive setting, and hypergraphs can be represented as graphs through clique/star expansions Agarwal et al. (2006). Note that while the number of edges changes due to these expansions, the number of nodes remains the same, which is why the proof holds.

### 3.2    USING CONTRASTIVE AUGMENTATIONS

While inefficiency quantifies a measure of the epistemic uncertainty, we have not yet accounted for the aleatoric uncertainty that can arise from a multitude of data-dependent properties. Minimizing the proxy of the epistemic uncertainty in isolation exposes our framework to noise that can arise from the structure of the hypergraph. As HGNNs rely on aggregating information by exploiting

the structural properties of hypergraphs, aleatoric uncertainties will be amplified by the model if unaccounted for. This motivated us to quantify and minimize the aleatoric uncertainty jointly with the epistemic uncertainty.

To execute this motivation, we utilize contrastive augmentations to boost the power of node embeddings in a self-supervised manner. We design contrastive structural augmentations akin to a prior work Wei et al. (2022) by constructing augmentations $\mathcal{H}_1 = \hat{f}(H, A_1)$ $\mathcal{H}_2 = \hat{f}(H, A_2)$ and corresponding node embeddings where $\hat{f}(\cdot, \cdot)$ is a function that perturbs the structure of a hypergraph given a perturbation schema $A$. Hence, $A_1$ and $A_2$ are two instantiations of the perturbation schema. Finally, we can obtain the node embeddings of the augmented hypergraphs as $\boldsymbol{Z}^1 = \tilde{\mu}(\mathcal{H}_1, \boldsymbol{X})$ $\boldsymbol{Z}^2 = \tilde{\mu}(\mathcal{H}_2, \boldsymbol{X})$ and minimizing the contrastive loss as follows:

$$\mathcal{L}_{\text{Contra}} = \text{InfoNCE}(\boldsymbol{Z}^1, \boldsymbol{Z}^2, \tau_2) = -\sum_{i=1}^{|\mathcal{V}|} \log \frac{\exp\left(\frac{\text{sim}(\mathbf{z}_i^1, \mathbf{z}_i^2)}{\tau_2}\right)}{\sum_{j=1}^{|\mathcal{V}|} \exp\left(\frac{\text{sim}(\mathbf{z}_i^1, \mathbf{z}_j^2)}{\tau_2}\right)}, \tag{2}$$

Here $\tau_2$ is a temperature hyperparameter to the popular InfoNCE loss Chen et al. (2020) and $\text{sim}(\cdot)$ denotes a similarity function like cosine similarity. The contrastive loss is also exchangeable as the loss depends on the embeddings, which are thus dependent on the mean estimator (HGNN in this case). As HGNN is permutation invariant, the contrastive loss is also exchangeable.

### 3.3 BOOSTING CONTRASTIVE AUGMENTATION WITH AUXILIARY HYPEREDGE DEGREE PREDICTION

To appropriately guide the calibration model $\tilde{\mu}(\cdot)$ with the structure of the hypergraph, we propose jointly training the hypergraph augmentations with the task of predicting the original hyperedge degrees. However, as the number of hyperedges in real-world hypergraphs is much greater than the number of nodes, we propose an efficient augmentation strategy to sample the most important hyperedges to perform the auxiliary hyperedge degree prediction task.

Let the hyperedge-Laplacian matrix of the hypergraph be $\boldsymbol{L} \in \mathbb{R}^{m \times n}$, where $m = |\mathcal{E}|$ is the number of hyperedges and $n = |\mathcal{V}|$ is the number of nodes. The hyperedge-Laplacian can be computed as $\boldsymbol{L} = \mathbf{D}_e^{-\frac{1}{2}} \mathbf{H}^T \mathbf{D}_v^{-\frac{1}{2}}$ Feng et al. (2019), where $\mathbf{H}$ denotes the incidence matrix. We apply self-attention mechanism Vaswani et al. (2017) over the hyperedge Laplacian to get attention weights $a_j = \text{Self-Attention}(\boldsymbol{L}_{:,j})$ for each hyperedge index $j$.

To sample the $k$ most important hyperedges in a fully differentiable manner, we use the Gumbel-Softmax trick Jang et al. (2016) as $\boldsymbol{s} = \text{GumbelSoftmax}(\boldsymbol{a}, k, \tau_3)$, where $\boldsymbol{s} \in \mathbb{R}^n$ is a soft selection mask, $k$ is the desired number of hyperedges, and $\tau_3$ is the temperature parameter. The auxiliary hyperedge degree prediction task is then $\hat{d}_j = h(\boldsymbol{L}_{:,j})$, where $h(\cdot)$ is a learnable predictor and $\hat{d}_j$ is the predicted degree of hyperedge $e_j$. Given the true degree $d_j$ for the hyperedge in the augmented hypergraph, the loss for the degree prediction task is $\mathcal{L}_{\text{deg}} = \sum_{j=1}^{n} s_j \cdot \ell(\hat{d}_j, d_j)$ where $\ell(\cdot, \cdot)$ is a regression loss, e.g., mean squared error. The degree prediction loss is also exchangeable as it does not relate to node labels in the transductive setting.

The overall training algorithm of our method is given in Algorithm 1 in the Appendix. This joint training encourages the model to learn representations sensitive to the structure of the most informative hyperedges while maintaining differentiability for end-to-end optimization.

### 3.4 THEORETICAL GUARANTEE

This section provides theoretical guarantees for our proposed method, in terms of shorter uncertainty band length (compared to the naive extension of the graph counterpart Huang et al. (2024) to hypergraphs). We will first define some notations that form the foundation of our theoretical results.

**Notations:** Assume an encoder-decoder architecture of the conformal corrector $\tilde{\mu}(\cdot)$, where the encoder maps the input node features to latent embeddings and the decoder maps those embeddings to predictions. Consider two models: (1) **CF-HGNN:** $\mathbf{Z}_0 = h_0(\mathcal{X})$ and $\hat{Y} = g_0(\mathbf{Z}_0)$ where $h_0(\cdot)$ and $g_0(\cdot)$ is the encoder and decoder, and $\mathbf{Z}_0$ is the latent representation. Its prediction set has expected

size $\mathcal{C}_0(\mathbf{x})$ given the node embedding $\mathbf{x}$. This is the naive extension of Huang et al. (2024) to hypergraphs. (2) **CCF-HGNN:** $\mathbf{Z}_1 = h_1(\mathcal{X}, A)$ and $\hat{Y}_1 = g_1(\mathbf{Z}_1)$ where $h_1(\cdot)$ and $g_1(\cdot)$ is encoder and decoder, and $\mathbf{Z}_1$ is the latent representation under contrastive augmentation $A$. Its prediction set has expected size $\mathcal{C}_1(\mathbf{x})$ given the node embedding $\mathbf{x}$. Recall, this is our proposed approach.

**Lemma 2.** *Let $I(Y; \mathbf{Z}_1)$ and $I(Y; \mathbf{Z}_0)$ denote the mutual information between the labels and latent embeddings for CCF-HGNN and CF-HGNN, respectively, and $\Delta \in \mathbf{R}^+$ then,*

$$I(Y; \mathbf{Z}_1) \geq I(Y; \mathbf{Z}_0) + \Delta. \tag{3}$$

The proof is provided in Appendix A.2. Using the results from Lemma 2, we can prove the following theorem on the expected band length produced by CCF-HGNN and CF-HGNN.

**Theorem 1.** *Under the assumptions:*

1. ***Bounded coverage:*** *Contrastive augmentations do not reduce conformal coverage (marginal coverage $\geq 1 - \alpha$ is preserved on average).*

2. ***Large Mutual Information gap:*** $I(Y; \mathbf{Z}_1) - I(Y; \mathbf{Z}_0)$ *is sufficiently large (Lemma 2).*

*Then, the expected conformal prediction set size under CCF-HGNN is smaller than under CF-HGNN:*

$$\mathbb{E}\big[|\mathcal{C}_1(\mathbf{x})|\big] \leq \mathbb{E}\big[|\mathcal{C}_0(\mathbf{x})|\big]. \tag{4}$$

The proof is provided in Appendix A.3. We also have a theoretical result on the band-length convergence guarantee for CCF-HGNN in the Appendix A.4..

## 4 EXPERIMENTS

Following the theoretical guarantees discussed earlier, we next demonstrate the empirical superiority of our proposed framework. Specifically, we evaluate the performance of our model and compare its performance against several non-trivial baselines on real-world datasets. We will first provide details about the experimental setup and then proceed to describe the evaluation metrics and experimental protocols, followed by the results.

### 4.1 SETUP

We conducted all experiments on AMD EPYC 7763 64-Core Processor with 1.08 TB memory and 8 NVIDIA A40 GPUs with CUDA version 13.0. Our code and experimental setup, including data construction, are available for peer review [1].

**Datasets:** We evaluated the performance of our proposed framework on four real-world datasets used in prior works Chien et al. (2021); Wang et al. (2023a). The datasets include co-authorship datasets like DBLP Yadati et al. (2019), co-purchases large dataset like Walmart-Trips Amburg et al. (2020), and co-voting datasets like House-Bills Chodrow et al. (2021), and Congress Fowler (2006). Summary statistics and further descriptions are provided in the Appendix A.5.

**Baseline Methods:** As there are no prior works tailored to quantify uncertainty for hypergraphs specifically, we use traditional uncertainty quantification methods (that do not provide statistical coverage guarantees) as baseline methods. These include Temperature Scaling (TS) Guo et al. (2017), Vector Scaling (VS) Guo et al. (2017), and Ensemble Temperature Scaling (ETS) Zhang et al. (2020). Additionally, we adapt traditional conformal prediction methods by adopting an HGNN mean estimator to obtain point predictions on hypergraphs (CP). Finally, we adapted the SOTA conformal prediction method for GNNs Huang et al. (2024) to aggregate information and perform conformal prediction in hypergraphs (CF-HGNN). Detailed descriptions of the baselines are provided in the Appendix A.6.

**Non-Conformity Score Functions:** We evaluate two popular conformal prediction scores.

---

[1]https://anonymous.4open.science/r/cont_conf_ml-3EB9

**(1) APS (Adaptive Prediction Sets) Romano et al. (2020):** For a model outputting class probabilities $\hat{p}(y \mid \mathbf{x})$, let $\pi(\mathbf{x})$ denote the ordering of labels sorted by decreasing probability. The APS score for class $y$ is defined as $V_{\text{APS}}(\mathbf{x}, y) = \sum_{j:\pi_j(\mathbf{x}) \prec y} \hat{p}(\pi_j(\mathbf{x}) \mid \mathbf{x}) + U \cdot \hat{p}(y \mid \mathbf{x})$, where $U \sim \text{Unif}(0, 1)$ and $\pi_j(\mathbf{x}) \prec y$ means label $\pi_j(\mathbf{x})$ is ranked higher than $y$. APS adaptively constructs prediction sets by accumulating probabilities until the threshold calibrated by conformal prediction is reached.

**(2) RAPS (Regularized Adaptive Prediction Sets) Angelopoulos et al. (2020):** RAPS extends APS by adding a regularization term that penalizes large set sizes. For class $y$, the score is $V_{\text{RAPS}}(\mathbf{x}, y) = S_{\text{APS}}(\mathbf{x}, y) + \lambda \cdot \big|\{j : \pi_j(\mathbf{x}) \prec y\}\big|^{\gamma}$, where $\lambda \geq 0$ controls the strength of the penalty and $\gamma \geq 1$ controls its growth rate. This modification encourages tighter prediction sets while preserving coverage guarantees.

**Evaluation Metrics:** We randomly split data into train, validation, calibration-test folds with a 20:30:50 split ratio. We adopt the following metrics to evaluate the empirical performance:

**(1) Marginal Coverage:** For a predictive confidence band $\mathcal{C}(\mathbf{x})$ and test point $(\mathbf{x}, y)$, the marginal coverage is defined as $\Pr\big(y \in \mathcal{C}(\mathbf{x})\big)$. A valid inference procedure should ensure that the empirical coverage satisfies $\Pr\big(y \in \mathcal{C}(\mathbf{x})\big) \geq 1 - \alpha$, where $\alpha$ is the target miscoverage rate.

**(2) Band Length:** Given that the empirical coverage exceeds $1 - \alpha$, the efficiency of the method is quantified by the expected length of the confidence band, $\mathbb{E}\big[\text{length}(\mathcal{C}(\mathbf{x}))\big]$. Comparisons of band length are only meaningful under the regime $\Pr\big(y \in \mathcal{C}(\mathbf{x})\big) \geq 1 - \alpha$, since trivially $\mathcal{C}(\mathbf{x}) = \emptyset$ yields zero length but violates the coverage constraint.

Table 2: Empirical Marginal Coverage (%) of different models for the task of node classification on four datasets with $\alpha = 0.05$. The result takes the average and standard deviation across 20 independent runs.

| Model | Walmart-Trips | House-Bills | Congress | DBLP | Covered? |
|---|---|---|---|---|---|
| TS | $92.26 \pm 0.31$ ✗ | $91.21 \pm 0.24$ ✗ | $89.04 \pm 0.48$ ✗ | $87.34 \pm 0.25$ ✗ | ✗ |
| VS | $92.20 \pm 0.18$ ✗ | $91.18 \pm 0.24$ ✗ | $88.99 \pm 0.46$ ✗ | $87.33 \pm 0.29$ ✗ | ✗ |
| ETS | $92.20 \pm 0.26$ ✗ | $92.93 \pm 1.77$ ✗ | $89.23 \pm 0.44$ ✗ | $88.29 \pm 0.65$ ✗ | ✗ |
| CP-APS | $95.17 \pm 0.00$ ✓ | $99.83 \pm 0.09$ ✓ | $99.61 \pm 0.02$ ✓ | $95.04 \pm 0.04$ ✓ | ✓ |
| CP-RAPS | $95.11 \pm 0.06$ ✓ | $95.20 \pm 0.04$ ✓ | $95.17 \pm 0.04$ ✓ | $95.13 \pm 0.03$ ✓ | ✓ |
| CF-HGNN-APS | $95.05 \pm 0.01$ ✓ | $99.97 \pm 0.00$ ✓ | $99.94 \pm 0.01$ ✓ | $97.31 \pm 2.58$ ✓ | ✓ |
| CF-HGNN-RAPS | $95.01 \pm 0.01$ ✓ | $95.18 \pm 0.10$ ✓ | $95.14 \pm 0.07$ ✓ | $95.07 \pm 0.01$ ✓ | ✓ |
| CCF-HGNN-APS (Ours) | $95.06 \pm 0.32$ ✓ | $99.68 \pm 0.00$ ✓ | $99.79 \pm 0.12$ ✓ | $99.49 \pm 0.39$ ✓ | ✓ |
| CCF-HGNN-RAPS (Ours) | $95.06 \pm 0.00$ ✓ | $95.33 \pm 0.03$ ✓ | $95.34 \pm 0.34$ ✓ | $95.06 \pm 0.04$ ✓ | ✓ |

## 4.2 Results

We will now provide empirical performances of all the baselines and our proposed framework to quantify uncertainty for classification tasks on the four datasets. The important conclusions derived from the experiments are listed below.

**All Conformal Frameworks Achieve the Desired Empirical Marginal Coverage while Traditional UQ Methods do not:** We report the marginal coverage of various UQ methods with target coverage at 95% in Table 2. There are two primary takeaways. Firstly, none of the traditional UQ methods (VS, TS, and ETS) achieves the target coverage for all datasets, while the conformal prediction methods (CP, CF-HGNN, and CCF-HGNN) do, highlighting the need for models with statistical guarantees when deployed in high-stakes environments. Secondly, these empirical results of all the conformal methods align with the theoretical coverage guarantee given in Lemma 1. Henceforth, we will only report the performance of models that obtain the desired coverage levels.

**Our Proposed Framework (CCF-HGNN) achieves the shortest Band Length in Most Datasets:** We report the empirical band length for 4 datasets in Table 3. The key observations are as follows. First, compared to standard conformal baselines (CP-APS, CP-RAPS). Our proposed approach CCF-HGNN-RAPS produces tighter bands across all but one dataset, while maintaining an impressive overall rank of 1.5 (the closest baselines get to 2.5). Second, while CF-HGNN offers improvements over GNN-based conformal methods, it is consistently outperformed by the proposed CCF-HGNN on hypergraph datasets. These results validate that incorporating contrastive learning with conformal prediction is crucial for boosting efficiency without compromising validity.

Table 3: Empirical Predictive Band Length of Different Models (that have the desired coverage level) on Four Datasets with $\alpha = 0.05$. The result takes the average and standard deviation across 20 independent runs. Lower is better.

| Model | Walmart-Trips | House-Bills | Congress | DBLP | Rank |
|---|---|---|---|---|---|
| CP-APS | $9.198 \pm 0.048$ | $1.958 \pm 0.005$ | $1.961 \pm 0.007$ | $3.479 \pm 0.127$ | 5.0 |
| CP-RAPS | $9.053 \pm 0.008$ | $\underline{1.261 \pm 0.054}$ | $\underline{1.317 \pm 0.007}$ | $\mathbf{1.509 \pm 0.038}$ | $\underline{2.5}$ |
| CF-HGNN-APS | $8.541 \pm 0.023$ | $1.993 \pm 0.013$ | $1.989 \pm 0.008$ | $4.346 \pm 0.387$ | 5.0 |
| CF-HGNN-RAPS | $8.595 \pm 0.400$ | $1.646 \pm 0.191$ | $1.619 \pm 0.129$ | $1.977 \pm 0.184$ | 3.25 |
| CCF-HGNN-APS (Ours) | $\mathbf{8.481 \pm 0.007}$ | $1.953 \pm 0.010$ | $1.949 \pm 0.008$ | $4.354 \pm 1.014$ | 3.75 |
| CCF-HGNN-RAPS (Ours) | $\underline{8.528 \pm 0.162}$ | $\mathbf{1.189 \pm 0.027}$ | $\mathbf{1.213 \pm 0.043}$ | $\underline{1.541 \pm 0.060}$ | $\mathbf{1.5}$ |

As observed in Table 2, APS-based conformal methods often produce empirical coverage well above the target level (close to $99\%$). This behavior arises because APS adaptively accumulates class probabilities until the calibration cutoff is exceeded, which in practice tends to overshoot the nominal threshold. While this conservativeness ensures validity, it also leads to overly large prediction sets. Consequently, APS methods trade efficiency for coverage, resulting in inflated band lengths (Table 3). By contrast, RAPS introduces an explicit penalty on the set size, thereby reducing redundancy in the prediction sets while still maintaining the desired coverage guarantees. However, Walmart-Trips is an exception as the difference between APS and RAPS is less pronounced, with APS achieving competitive band lengths relative to RAPS. This can be attributed to the nature of Walmart-Trips, which has a relatively large number of classes (11) but moderate class imbalance. In such settings, APS's conservative accumulation of probabilities does not inflate the prediction sets as severely as in smaller-class datasets, since the distribution of probabilities is already more spread out across labels. As a result, while RAPS still improves efficiency, the margin of improvement over APS is narrower on Walmart-Trips compared to the other datasets.

**Ablation Study:** We analyze the effect of removing three key components—the *topological-aware conformal loss*, *auxillary degree prediction loss*, and *contrastive loss*—on the **Congress** and **House-Bills** datasets on at a time. Figure 2 reports coverage and band length under RAPS with $\alpha = 0.05$. Our key observations are: (1) *Topology-aware conformal prediction loss is crucial*: removing it inflates RAPS length substantially (e.g., $1.213 \to 1.469$ on Congress, $1.214 \to 1.744$ on House-Bills), showing that structural

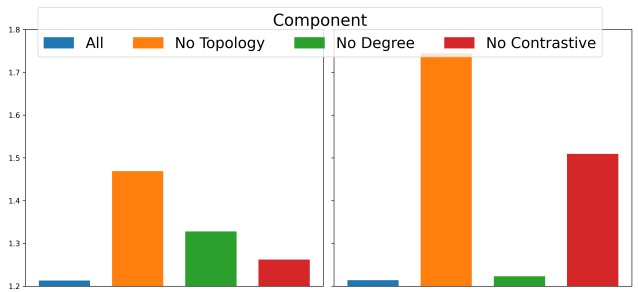

Figure 2: Ablation Study: Variation band length (right) for RAPS on CCF-HGNN on Congress (left) and House-Bills (right) dataset due to removal of individual components for $\alpha = 0.05$. Smaller is better.

information yields tighter sets. (2) *Minimizing the auxillary loss helps*: excluding degree modestly increases lengths (e.g., $1.213 \to 1.328$ on Congress). (3) *Contrastive learning improves efficiency*: dropping it slightly lengthens sets (e.g., $1.213 \to 1.262$ on Congress). Overall, each component contributes to efficiency, with topology offering the largest gains. The complete model yields the tightest bands while maintaining the desired coverage guarantees.

Table 4: Effect of different contrastive strategies (mean $\pm$ std dev across 20 runs.) for $\alpha = 0.05$.

| Dataset | Technique | APS Coverage | APS Length | RAPS Coverage | RAPS Length |
|---|---|---|---|---|---|
| Congress | Hyperedge Drop | $99.83 \pm 0.13$ | $1.997 \pm 0.011$ | $95.27 \pm 0.19$ | $1.309 \pm 0.046$ |
| | Edge Drop | $99.79 \pm 0.12$ | $1.949 \pm 0.008$ | $95.34 \pm 0.034$ | $1.213 \pm 0.043$ |
| DBLP | Hyperedge Drop | $99.39 \pm 0.10$ | $3.688 \pm 0.384$ | $95.08 \pm 0.16$ | $1.641 \pm 0.132$ |
| | Edge Drop | $99.49 \pm 0.39$ | $4.354 \pm 1.014$ | $95.06 \pm 0.04$ | $1.541 \pm 0.060$ |
| House-Bills | Hyperedge Drop | $99.68 \pm 0.00$ | $1.953 \pm 0.010$ | $95.33 \pm 0.03$ | $1.189 \pm 0.027$ |
| | Edge Drop | $99.64 \pm 0.03$ | $1.955 \pm 0.006$ | $95.19 \pm 0.00$ | $1.214 \pm 0.018$ |
| Walmart-Trips | Hyperedge Drop | $95.05 \pm 0.05$ | $8.506 \pm 0.136$ | $95.05 \pm 0.00$ | $8.571 \pm 0.102$ |
| | Edge Drop | $95.06 \pm 0.32$ | $8.481 \pm 0.007$ | $95.06 \pm 0.00$ | $8.528 \pm 0.162$ |

**Sensitivity Study 1: Different Augmentation Strategies** To account for aleatoric uncertainty, we exploit contrastive augmentations by perturbing the hypergraph structure. We compare two

strategies: (i) *random hyperedge drop*, which removes entire hyperedges, and (ii) *random edge drop*, which removes individual edges in the bipartite node–hyperedge graph. Table 4 summarizes the results. Across datasets, both strategies achieve the target coverage, but their impact on efficiency differs. On **Congress** and **House-Bills**, edge drop consistently yields shorter RAPS sets (e.g., 1.213 vs. 1.309 on Congress), indicating that fine-grained perturbations help the model learn more stable and discriminative representations. In contrast, **DBLP** benefits slightly more from hyperedge drop, where APS sets are tighter (3.688 vs. 4.354), suggesting that larger-scale perturbations are useful in high-homophily graphs with many small hyperedges. For **Walmart-Trips**, the differences between the two strategies are marginal, likely due to its large number of classes and moderate imbalance, where both perturbations introduce comparable variability. Overall, edge drop is generally more effective for heterophilic co-voting datasets, while hyperedge drop can be advantageous for homophilic graphs like DBLP. This demonstrates the importance of tailoring contrastive augmentation strategies to the structural properties of the underlying hypergraph.

**Sensitivity Study 2: Dependence on Confidence Level** We further study the sensitivity of our method to two key parameters: the miscoverage rate $\alpha$ (i.e., target confidence level) and the calibration set size. Figure 3 shows the results of this experiment for **Congress** and **House-Bills** datasets. Figure 3a and Figure 3b show the change in predictive band length as the confidence level increases from 0.7 to 0.95. Across both datasets, the band length grows monotonically with confidence, as expected. While all methods follow this trend, our method consistently achieves shorter band lengths compared to CP and CF-HGNN, especially at higher confidence levels (e.g., $\alpha = 0.05$). This demonstrates that our contrastive framework yields more informative uncertainty estimates without sacrificing coverage.

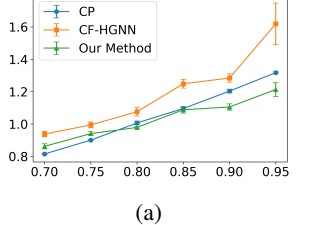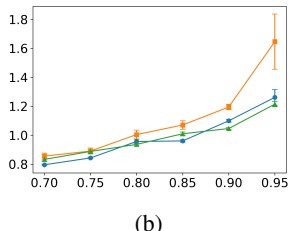

(a)                             (b)

Figure 3: Sensitivity study on varying $\alpha$ for Congress (3a) and House-Bills (3b).

**Sensitivity Study 3: Size of Calibration Set** We also evaluate the effect of calibration set fraction (25%, 50%, 75%). Results in Figure 4a and Figure 4b show that our method remains stable with minimal fluctuation in band length as calibration data decreases. In contrast, CF-HGNN exhibits higher variance and inflated intervals, especially at smaller calibration fractions. This stability highlights the robustness of our approach under limited calibration resources, which is important in real-world healthcare applications where labeled calibration data may be scarce.

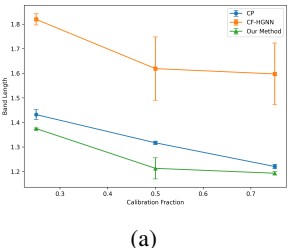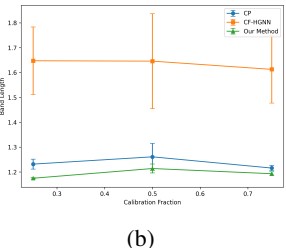

(a)                             (b)

Figure 4: Sensitivity Study on varying the calibration set fraction (4a and 4b) for Congress and House-Bills datasets respectively.

**Performance in Multi-Class Hypergraph Datasets** To additionally observe the performance of the conformal prediction methods for multi-class datasets, we used two datasets, namely **Trivago-Clicks** and **High-School**. The descriptions and summaries of these datasets are provided in Section A.5.

Table 5: APS and RAPS coverage and length across multi-class datasets for $\alpha = 0.05$. Mean $\pm$ standard deviation over 20 runs.

| Dataset | Model | APS Coverage | APS Length | RAPS Coverage | RAPS Length |
|---|---|---|---|---|---|
| **High-School** | CP | $97.86 \pm 0.45$ | $7.81 \pm 0.07$ | $96.14 \pm 0.00$ | $7.66 \pm 0.04$ |
| | CF-HGNN | $97.33 \pm 0.01$ | $7.44 \pm 0.06$ | $95.98 \pm 0.06$ | $7.28 \pm 0.05$ |
| | Ours | $97.79 \pm 0.00$ | $7.30 \pm 0.32$ | $96.13 \pm 0.00$ | $\mathbf{7.13 \pm 0.73}$ |
| **Trivago-clicks** | CP | $95.08 \pm 0.04$ | $56.23 \pm 2.55$ | $95.11 \pm 0.04$ | $54.79 \pm 1.41$ |
| | CF-HGNN | $95.03 \pm 0.00$ | $52.09 \pm 1.04$ | $95.01 \pm 0.03$ | $51.38 \pm 2.01$ |
| | Ours | $95.11 \pm 0.04$ | $51.47 \pm 1.44$ | $95.11 \pm 0.00$ | $\mathbf{50.89 \pm 1.66}$ |

The results in Table 5 show that all conformal methods maintain the desired marginal coverage at the target level $\alpha = 0.05$. However, consistent with our observations in the main paper, our proposed approach achieves notably shorter prediction sets—particularly under RAPS, demonstrating improved efficiency while preserving valid coverage. On both datasets, our method outperforms CP and CF-HGNN in terms of band length, with the largest gains observed on **Trivago-Clicks**, where the high number of classes amplifies the benefits of contrastive regularization and topology-aware conformal correction. These results further confirm that jointly modeling aleatoric and epistemic uncertainties yields tighter and more informative prediction sets in multi-class hypergraph settings.

## 5 RELATED WORKS

In this section, we briefly discuss some important works that have not been discussed before. For a more comprehensive survey, refer to Appendix A.7.

(1) Uncertainty Quantification (UQ) on Networks: Traditional UQ methods on graph have gained more popularity over time Zhao et al. (2020); Stadler et al. (2021); Bertozzi et al. (2018); Han et al. (2025); Srinivasan et al. (2018) that has influenced training strategies Kang et al. (2022); Trivedi et al. (2024a) and other applications Huang & Chung (2020); Yu et al. (2024). While some methods have been proposed for hypergraphs Yao et al. (2025); Harit & Sun (2025), they are focused towards applications and not generalizable.

(2) Conformal Prediction: Due to the statistical guarantee and distribution-free assumptions, conformal prediction has become very popular in recent times. Some directions include conditional conformal prediction Ding et al. (2023); Gibbs et al. (2025); Luo & Zhou (2025), reformulation of conformal prediction to other domains Correia et al. (2024); Cherian et al. (2024) and conformal prediction under distribution shift Barber et al. (2023); Clarkson (2023); Thopalli et al. (2025).

## 6 CONCLUSION

In this work, we extend the notion of UQ on hypergraphs by jointly accounting for both aleatoric and epistemic sources of uncertainty and proposing a hypergraph-based conformal prediction framework that leads to improved band lengths. While this is a promising direction, potential directions of future work include the evaluation of the performance of other HGNN models like Allset Chien et al. (2021), ED-HNN Wang et al. (2023a), and accounting for other sources of aleatoric uncertainty. On the side of conformal prediction, possible future directions include evaluation in the inductive setting Zargarbashi & Bojchevski (2023); Clarkson (2023) where the assumption of exchangeability is not maintained.

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

## 7 LLM USAGE AND ETHICS STATEMENT

This paper presents work whose goal is to advance the field of Representation Learning. There are many potential societal consequences of our work, none of which we feel must be specifically highlighted here. LLMs were used to correct grammatical errors that originated during the writing of the manuscript. LLMs were not used to create any ideas. Our source code and experimental setup are provided through the anonymised link in the manuscript. All the datasets used in this work are publicly available, and there are no competing interests.

## A APPENDIX

### A.1 HYPERGRAPH NEURAL NETWORKS

The early structure of HGNNs mimicked the convolution step of GNNs. In particular, Feng et al. Feng et al. (2019) proposed the first spectral hypergraph convolution, formulated as

$$\mathbf{X}' = \sigma\left(\mathbf{D}_v^{-\frac{1}{2}}\mathbf{H}\mathbf{W}_e\mathbf{D}_e^{-1}\mathbf{H}^\top\mathbf{D}_v^{-\frac{1}{2}}\mathbf{X}\mathbf{W}\right), \tag{5}$$

where $\mathbf{H}$ is the incidence matrix, $\mathbf{D}_v$ and $\mathbf{D}_e$ are vertex and hyperedge degree matrices, $\mathbf{W}_e$ is a diagonal hyperedge weight matrix, and $\mathbf{W}$ is a trainable weight matrix.

Later, Bai et al. (2021) introduced a simplified hypergraph convolution operation, expressed as

$$\mathbf{X}' = \sigma\left(\mathbf{D}_v^{-1}\mathbf{H}\mathbf{D}_e^{-1}\mathbf{H}^\top\mathbf{X}\mathbf{W}\right), \tag{6}$$

which removes the symmetric normalization and leads to a message-passing view of hypergraph learning. This formulation laid the foundation for subsequent works such as UniGNN Huang & Yang (2021). In all our experiments, we have used the formulation by Bai et al. (2021).

### A.2 PROOF OF LEMMA 2

*Proof.* From Proposition E.2 in Oord et al. (2018), we know

$$I(Y; \mathbf{Z}) \geq \log(N) - \mathcal{L}_{\mathbf{Z}}^{\text{InfoNCE}},$$

where $N$ is the number of samples and $\mathcal{L}_{\mathbf{Z}}^{\text{InfoNCE}}$ the InfoNCE loss.

For our method, $\mathcal{L}_{\mathbf{Z}_1}^{\text{InfoNCE}}$ is explicitly minimized, compared to CF-HGNN, which consequently means that $\log(N) - \mathcal{L}_{\mathbf{Z}_1}^{\text{InfoNCE}} \geq \log(N) - \mathcal{L}_{\mathbf{Z}_0}^{\text{InfoNCE}}$.

This implies the proof statement:

$$I(Y; \mathbf{Z}_1) \geq I(Y; \mathbf{Z}_0) + \Delta$$

$\square$

### A.3 PROOF OF THEOREM 1

**Lemma 3.** *Correia et al. (2024) For any conformal prediction scheme with the coverage guarantee of $1 - \alpha$, and any distribution $q(\cdot)$, we have:*

$$\mathbb{E}([\log |\mathcal{C}(x)|]^+) \geq$$

$$(1-\alpha)\frac{H(Y|X) - h_b(a) - a\log M - \alpha\mathbb{E}_{P_{Y,X,\mathcal{D}_{cal}|E=0}}\left[-\log\hat{Q}_{Y|X}^0 + \log\mathbb{E}_{u(y_{\mathcal{C}(\bar{x})})}[q(y|x)]\right]}{1 - \alpha + \frac{1}{n+1}}$$

$$- (1-\alpha)\mathbb{E}_{P_{Y,X,\mathcal{D}_{cal}|E=1}}\left[-\log\hat{Q}_{Y|X}^1 + \log\mathbb{E}_{u(y_{\mathcal{C}(x)})}[q(y|x)]\right], \tag{7}$$

*where $\hat{Q}_{Y|X}^0 = q(y|x)\mathbb{I}[y \notin \mathcal{C}(x)]$ and $\hat{Q}_{Y|X}^1 = q(y|x)\mathbb{I}[y \in \mathcal{C}(x)]$. Here, $|\mathcal{C}(x)|$ denotes the size of the prediction set for input $x$, $H(Y|X)$ the conditional entropy of $Y$ given $X$, $h_b(\cdot)$ the binary entropy function, $a$ the error probability, and $M = |\mathcal{Y}|$ the number of classes.*

---

**Algorithm 1** Contrastive Hypergraph Conformal Prediction (CCF-HGNN)

---

**Input:** Hypergraph $H = \{\mathcal{V}, \mathcal{E}\}$, feature matrix $\boldsymbol{X}$, label set $\mathcal{Y}$, Incidence Matrix $H$
HGNN train Model $\hat{\mu}(\cdot)$, calibration model $\tilde{\mu}(\cdot)$ with weights $\vartheta$, non-conformity score function
$V(\cdot, \cdot)$, Calibration dataset $\mathcal{D}_{\text{cal}}$ partitioned into $\mathcal{D}_{\text{corr-cal}} = \{(\mathbf{x}_i, y_i)\}_{i=1}^{n_{\text{corr-cal}}}$,
and $\mathcal{D}_{\text{cal-test}} = \{(\boldsymbol{x}_i, y_i)\}_{i=1}^{n_{\text{cal-test}}}$, significance level $\alpha$, Hypergraph incidence, node and hyperedge
degree matrices $\mathbf{H}, \mathbf{D}_e, \mathbf{D}_v$
1: Train HGNN model $\hat{\mu}(H, \boldsymbol{X})$ on prediction task.
2: **while** Not converged **do**
3:     Obtain augmentations $\mathcal{H}_1$ and $\mathcal{H}_2$ of $H$.
4:     Compute the hyperedge Laplacians $\boldsymbol{L}_1$ and $\boldsymbol{L}_2$, corresponding attention weights $\boldsymbol{a}^1$ and $\boldsymbol{a}^2$.
5:     Select the k important hyperedges using the Gumbel-Softmax trick.
6:     Compute the overall degree loss $\mathcal{L}_{\text{deg}} = \mathcal{L}_{\text{deg}}^1 + \mathcal{L}_{\text{deg}}^2$.
7:     Get embeddings $\mathbf{Z}^1 = \tilde{\mu}(\mathcal{H}_1, f(\boldsymbol{X})), \mathbf{Z}^2 = g(\mathcal{H}_2, f(\boldsymbol{X}))$.
8:     Get calibration predictions $\mathbf{Z}_{\text{cal}}^1, \mathbf{Z}_{\text{cal}}^2$ from $\mathbf{Z}^1, \mathbf{Z}^2$.
9:     Compute $\mathbf{Z}_{\text{cal}} = \frac{\mathbf{Z}_{\text{cal}}^1 + \mathbf{Z}_{\text{cal}}^2}{2}$.
10:    Get test predictions $\mathbf{Z}_{\text{test}}^1, \mathbf{Z}_{\text{test}}^2$ from $\mathbf{Z}^1, \mathbf{Z}^2$.
11:    Compute $\mathbf{Z}_{\text{test}} = \frac{\mathbf{Z}_{\text{test}}^1 + \mathbf{Z}_{\text{test}}^2}{2}$.
12:    Compute $\hat{\alpha} = \frac{1}{n+1} \cdot \alpha$.
13:    $\hat{\eta} = \text{DiffQuantile}(\{V(\mathbf{Z}_i, y_i) \mid i \in \mathcal{D}_{\text{cal}}\})$.
14:    $\mathcal{L}_{\text{Ineff}} = \frac{1}{m} \sum_{i \in \mathcal{D}_{\text{cal-test}}} \frac{1}{|\mathcal{Y}|} \sum_{k \in \mathcal{Y}} \sigma \left( \frac{V(\mathbf{z}_i, k) - \hat{\eta}}{\tau_1} \right)$.
15:    $\mathcal{L}_{\text{Contra}} = \text{INFONCE}(\mathbf{Z}^1, \mathbf{Z}^2, \tau_2)$
16:    $\mathcal{L}_{\text{Total}} = \gamma \mathcal{L}_{\text{Ineff}} + (1 - \gamma) \mathcal{L}_{\text{Contra}} + \mathcal{L}_{\text{deg}}$.
17:    $\vartheta = \vartheta - \nabla_\vartheta \mathcal{L}_{\text{Total}}$.
18: **end while**

---

Lemma 3 shows that the expected prediction set size is lower bounded by the conditional entropy $H(Y|X)$, penalized by calibration-dependent terms.

*Proof.* By Lemma 2, $I(Y; Z_1) \geq I(Y; Z_0) + \Delta$. Equivalently, $H(Y|Z_1) \leq H(Y|Z_0) - \Delta$.

Lemma 3 lower bounds the expected log set size in terms of $H(Y|X)$. Since $Z_1$ captures more information about $Y$ than $Z_0$, the effective conditional entropy $H(Y|Z_1)$ is smaller. Thus, the bound for $\mathcal{C}_1(X)$ is tighter than for $\mathcal{C}_0(X)$.

Formally,

$$\mathbb{E}[\log |\mathcal{C}_0(X)||]^+ \geq f(H(Y|Z_0)),$$
$$\mathbb{E}[\log |\mathcal{C}_1(X)||]^+ \geq f(H(Y|Z_1)),$$

where $f(\cdot)$ is the lower-bound functional in Lemma 3. Since $H(Y|Z_1) < H(Y|Z_0)$, the bound for $\mathcal{C}_1(X)$ is strictly smaller, which implies:

$$\mathbb{E}[|\mathcal{C}_1(X)|] \leq \mathbb{E}[|\mathcal{C}_0(X)|].$$

$\square$

## A.4 CONVERGENCE OF CCF-HGNN

**Theorem 2.** *If the calibration model $\tilde{\mu}(\cdot)$ produces stable predictions $\hat{p}(y_i|\mathcal{X}_i)$ as the number of calibration samples $n_{cal} \to \infty$, the expected prediction set size $\mathbb{E}[|C(\mathbf{x})|]$ for a test point converges in probability to a fixed value:*

$$\mathbb{E}[|C(\mathbf{x})|] \to \sum_{y \in \mathcal{Y}} \mathbb{P}(\hat{p}(y|\mathbf{x}) \geq 1 - q^*), \tag{8}$$

*where $q^* = F^{-1}(1 - \alpha)$ is the $(1 - \alpha)^{th}$-quantile of the true non-conformity score distribution.*

*Proof.* Let $F_n(v) = \frac{1}{n} \sum_{i=1}^{n} \mathbf{1}(V_i \leq v)$ be the empirical CDF of the non-conformity scores $V_i = 1 - \hat{p}(y_i|\mathcal{X}_i)$.

Using Glivenko-Cantelli Theorem Van der Vaart (2000) with Assumption 1, $\sup_v |F_n(v) - F(v)| \to 0$ as $n \to \infty$. Assuming the calibration model $\tilde{\mu}(\cdot)$ produces stable predictions $\hat{p}(y_i|\mathcal{X}_i)$ as the number of calibration samples $n_{cal} \to \infty$, and as $F$ is continuous and strictly increasing, $F^{-1}$ is continuous at $1 - \alpha$.

For any $\epsilon > 0$, choose $\delta > 0$ such that

$$F(q^* - \epsilon) < 1 - \alpha - \delta, \quad F(q^* + \epsilon) > 1 - \alpha + \delta.$$

As $n_{cal}$ grows, $\sup_v |F_{n_{cal}}(v) - F(v)| < \delta$, which means

$$F_{n_{cal}}(q^* - \epsilon) \geq F(q^* - \epsilon) - \delta < 1 - \alpha,$$

and

$$F_{n_{cal}}(q^* + \epsilon) \leq F(q^* + \epsilon) + \delta > 1 - \alpha.$$

So $q^* - \epsilon < \hat{q} < q^* + \epsilon$, which means

$$\mathbb{P}(|\hat{q} - q^*| > \epsilon) \to 0 \quad \text{as } n_{cal} \to \infty.$$

The prediction set is thus

$$C(\mathbf{x}) = \{y \in \mathcal{Y} : \hat{p}(y|\mathbf{x}) \geq 1 - \hat{q}\}.$$

So the expected set size is

$$\mathbb{E}[|C(\mathbf{x})|] = \mathbb{E}\left[\sum_{y \in \mathcal{Y}} \mathbf{1}(\hat{p}(y|\mathbf{x}) \geq 1 - q)\right] = \sum_{y \in \mathcal{Y}} \mathbb{P}(\hat{p}(y|\mathbf{x}) \geq 1 - q).$$

As $\hat{q} \to q^*$, and since $g(\cdot)$ is stable,

$$\mathbf{1}(\hat{p}(y|\mathbf{x}) \geq 1 - \hat{q}) \to \mathbf{1}(\hat{p}(y|\mathbf{x}) \geq 1 - q^*).$$

So,

$$\mathbb{E}[|C(\mathbf{x})|] \to \sum_{y \in \mathcal{Y}} \mathbb{P}(\hat{p}(y|\mathbf{x}) \geq 1 - q^*).$$

This limit is a fixed value determined by the distribution of $\hat{p}(y|\mathbf{x})$ and $q^*$.

Conformal prediction ensures that as long as $\hat{q}$ is calibrated,

$$\mathbb{P}(y \in C(\mathbf{x})) \geq 1 - \alpha.$$

$\square$

## A.5 DESCRIPTIONS OF THE DATASETS

Table 6: Statistics of the selected datasets. Here, DBLP is a homophilic dataset while the others are heterophilic.

| Property | DBLP | Congress | House-Bills | Walmart-Trips | High-School | Trivago-Clicks |
|---|---|---|---|---|---|---|
| # nodes | 41,302 | 1,718 | 1,494 | 88,860 | 327 | 170,994 |
| # hyperedges | 22,363 | 83,105 | 60,987 | 69,906 | 7818 | 232,013 |
| # classes | 6 | 2 | 2 | 11 | 9 | 80 |
| avg. $|e|$ | 4.452 | 8.656 | 20.500 | 6.589 | 2.300 | 3.116 |

This work uses four hypergraph classification datasets in the main text. They are as follows:

- **Walmart-Trips:** This is a customer recruitment prediction dataset where the hyperedges are sets of co-purchased products at Walmart. Products (nodes) are assigned to one of ten broad departments in which the product appears on walmart.com (e.g., "Clothing, Shoes, and Accessories"), and these serve as node labels (there is also an additional "Other" class).

- **DBLP:** This is a co-authorship hypergraph dataset created by Yadati et al. (2019). It represents collaborations among authors listed in DBLP, the computer science bibliographic database, as of 3 Sept. 2017. Each node represents an author, and each publication is represented by a simplex (a set of nodes, i.e., a hyperedge), timestamped by the year of publication. This is the only homophilic hypergraph dataset

- **Congress:** In this hypergraph dataset, nodes are US Congresspersons and simplices are comprised of the sponsor and co-sponsors of legislative bills put forth in both the House of Representatives and the Senate.

- **House-Bills:** In this hypergraph dataset, nodes are US Congresspersons and hyperedges are the sponsors and co-sponsors of bills put forth in the House of Representatives. Some hyperedges are repeated. Each node is labeled with political party affiliation.

Additionally, this work also used two multi-class hypergraph datasets. They are as follows:

- **High-School** Chodrow et al. (2021); Mastrandrea et al. (2015): This is a static, annotated hypergraph version of the temporal higher-order contact-high-school dataset. Each hyperedge corresponds to a group of people who were all in proximity of one another at a given time, based on data from sensors worn by students. Each node is labeled with the classroom to which the student belongs.

- **Trivago-Clicks** Chodrow et al. (2021): This is a hypergraph, where nodes are accommodations (mostly hotels), and hyperedges are sets of accommodations for which a user performed the "click-out" action during the same browsing session, which means the user was forwarded to a partner site. Although the original dataset has 160 node classes, a lot of them are singular. We selected the nodes belonging to the top 80 labels in our experiments.

### A.6 DESCRIPTIONS OF THE BASELINES

The baseline models used in this work can be characterized into the following categories:

- **Traditional UQ Methods:** These methods do not provide any statistical guarantee about marginal coverage. The 3 baseline methods used under this category are as follows:

  1. **Temperature Scaling (TS) Guo et al. (2017):** It is a post-processing calibration method for UQ. It takes the model's logits (pre-softmax outputs) and divides them by a learned scalar parameter called the temperature. Higher temperature values produce softer probability distributions with lower confidence.
  2. **Vector Scaling (VS) Guo et al. (2017):** Vector scaling is a more flexible version of temperature scaling. Instead of using a single global adjustment for all classes, it assigns each class its own adjustment with a small bias. This allows the model to adjust situations where some classes are consistently overconfident or underconfident, thereby improving the calibration of predicted probabilities across all classes.
  3. **Ensemble Temperature Scaling (ETS) Zhang et al. (2020):** Ensemble Temperature Scaling applies temperature scaling to the aggregated outputs of a model ensemble. A single temperature parameter is learned on the ensemble's averaged logits to adjust overall confidence. This method preserves the accuracy advantages of ensembling while improving calibration, resulting in more reliable uncertainty estimates.

- **Conformal Prediction Methods:** These methods have a theoretical guarantee for marginal coverage. We adapted two prior works as baselines:

  1. **Conformal Predictor (CP) Vovk et al. (2005):** For this model, the mean estimator (HGNN) was trained on the classification task on the training data. After that, the non-conformity scores were obtained for the calibration data (node set), a quantile was selected (based on the type of the non-conformity score function), and predictive bands were constructed for test nodes.
  2. **Conformalized Hypergraph Neural Network Huang et al. (2024) (CF-HGNN):** This model integrates conformal prediction with hypergraph neural networks to provide uncertainty estimates with guaranteed marginal coverage. The key idea is to adapt non-conformity scores to hypergraph learning tasks, where nodes, edges, and higher-order relationships need to be considered simultaneously. CF-HGNN first trains a base

HGNN to produce class probability estimates, then applies a conformal calibration step using a held-out calibration set. Unlike CP, CF-HGNN explicitly accounts for hypergraph structures, leading to tighter predictive sets and better utilization of higher-order relational information. As such, it represents the current state-of-the-art approach for principled uncertainty quantification in hypergraph datasets, balancing theoretical guarantees with strong empirical performance.

## A.7 RELATED WORKS

We discuss here related works that are closest to the ideas in CCF-HGNN in this section.

(1) Uncertainty Quantification in deep learning and GNNs: Several approaches address model-agnostic risk estimation for Graph Neural Networks (GNNs) in both classification and regression tasks Zhang et al. (2020); Ovadia et al. (2019); Seedat et al. (2022). Other studies leverage structural properties of graphs to explore calibration challenges, particularly the tendency of GNNs to be underconfident Wang et al. (2021); Hsu et al. (2022). A foundational perspective is provided by Gal & Ghahramani (2016), who interpret dropout training in deep neural networks as approximate Bayesian inference in deep Gaussian Processes. Complementary work investigates factors such as network depth, width, weight decay, batch normalization, and temperature scaling for improving calibration Lakshminarayanan et al. (2017); Guo et al. (2017). More recently, stochastic centering has been proposed and applied as an effective calibration technique for GNNs Trivedi et al. (2023; 2024b).

(2) Conformal Prediction: Conformal inference provides distribution-free uncertainty quantification with rigorous coverage guarantees, enabling applications across diverse domains such as model calibration Sweidan & Johansson (2021), passenger booking systems Werner et al. (2021), computer vision Angelopoulos et al. (2020); Bates et al. (2021), and time-series forecasting Gibbs & Candes (2021); Lin et al. (2022). Given a user-specified miscoverage rate $\alpha \in (0, 1)$, the framework uses a calibration dataset to construct prediction sets or intervals that contain the true outcome with probability at least $1 - \alpha$. A variety of nonconformity scores have been proposed to improve performance in classification settings Romano et al. (2019; 2020); Izbicki et al. (2019), with recent work introducing scores in the latent feature space Teng et al. (2022). While the classical framework relies on exchangeability, several extensions relax this assumption to handle label shift, covariate shift, or dependent data Gibbs & Candes (2021); Barber et al. (2023); Tibshirani et al. (2019); Lin et al. (2022).

(3) Conformal Prediction for GNNs: The use of conformal inference for network-structured data has recently gained traction. The first application in the inductive setting Clarkson (2023) demonstrated that nonconformity scores in this context are not exchangeable. In contrast, subsequent works Huang et al. (2024); Zargarbashi et al. (2023); Luo & Colombo (2024) study the transductive setting, where nonconformity scores retain exchangeability. These approaches exploit the local neighborhood structure of graphs to improve effectiveness while maintaining computational efficiency. More recently, Zargarbashi & Bojchevski (2023) introduced the notions of node-exchangeability and edge-exchangeability in growing graphs for the inductive setting, and proposed nonconformity scores defined on the evolving graph structure at each step. Recent works also include conformalized link prediction Zhao et al. (2024), weighted edge prediction Choudhuri et al. (2025c); Luo & Colombo (2024), dynamic GNNs Davis et al. (2024); Wang et al. (2025) and adversasial attack detection Ennadir et al. (2023).

## A.8 ABLATION STUDY: OPTIMIZING OTHER NON-CONFORMITY SCORES

While our main experimental results were based on optimizing APS, we performed an additional experiment by using DAPS Zargarbashi et al. (2023) as the non-conformity score function. The neighbour diffused scores of DAPS is given by $\hat{\mathbf{H}} = (1 - \lambda)\mathbf{H} + \mathbf{D}^{-1}\mathbf{A}\mathbf{H}$, where $\mathbf{D}$ denotes the node degree matrix, $\mathbf{A}$ denotes the node adjacency matrix and $\mathbf{H}$ denotes the node-wise score matrix. We experimented on **Walmart-Trips** and **DBLP** datasets for a target coverage of $95\%$. The results of our experiments are presented in Table 7.

We notice that for **Walmart-Trips**, using DAPS does not improve performance for our method compared to optimizing the APS score in Table 3. However, the performance of our method improves

Table 7: Average Marginal Coverage and Band Length for all conformal methods using DAPS non-conformity score across 20 runs. $\alpha = 0.05$

| Dataset | Method | Coverage | Length |
|---|---|---|---|
| Walmart-Trips | CP-APS | $95.16 \pm 0.02$ | $8.973 \pm 0.003$ |
| | CP-RAPS | $95.08 \pm 0.01$ | $8.901 \pm 0.053$ |
| | CF-HGNN-APS | $95.05 \pm 0.02$ | $\underline{8.588 \pm 0.003}$ |
| | CF-HGNN-RAPS | $95.01 \pm 0.00$ | $8.611 \pm 0.001$ |
| | Our-APS | $95.07 \pm 0.03$ | $\mathbf{8.518 \pm 0.001}$ |
| | Our-RAPS | $95.05 \pm 0.01$ | $8.592 \pm 0.003$ |
| DBLP | CP-APS | $97.05 \pm 0.00$ | $3.470 \pm 0.065$ |
| | CP-RAPS | $95.07 \pm 0.00$ | $\underline{1.600 \pm 0.011}$ |
| | CF-HGNN-APS | $97.11 \pm 0.01$ | $3.920 \pm 0.011$ |
| | CF-HGNN-RAPS | $95.08 \pm 0.00$ | $1.703 \pm 0.002$ |
| | Our-APS | $98.79 \pm 0.01$ | $3.734 \pm 0.067$ |
| | Our-RAPS | $95.13 \pm 0.00$ | $\mathbf{1.508 \pm 0.067}$ |

when evaluated on the **DBLP dataset**. This is primarily due to the fact that the non-conformity score of DAPS induces homophily and thus does not improve performance in a heterophilous hypergraph like **Walmart-Trips**. However, for a homophilous hypergraph like **DBLP**, the performance of all methods improves significantly when using an appropriate non-conformity score.

## A.9 EFFECT OF MEAN ESTIMATOR ON OUR METHOD

Table 8: Performance of Models using ED-HNN averaged across 20 runs. $\alpha = 0.05$

| Dataset | Model | APS Coverage | APS Length | RAPS Coverage | RAPS Length |
|---|---|---|---|---|---|
| Congress | CP | $99.49 \pm 0.00$ | $1.95 \pm 0.00$ | $95.29 \pm 0.00$ | $1.78 \pm 0.12$ |
| | Ours | $98.99 \pm 0.00$ | $1.97 \pm 0.02$ | $95.28 \pm 0.00$ | $1.40 \pm 0.05$ |
| House-Bills | CP | $99.53 \pm 0.00$ | $1.95 \pm 0.01$ | $95.25 \pm 0.00$ | $1.24 \pm 0.07$ |
| | Ours | $98.74 \pm 0.01$ | $1.96 \pm 0.03$ | $95.19 \pm 0.00$ | $1.15 \pm 0.15$ |

The mean estimator used for all experiments in the main text was HCHA Bai et al. (2021). As the conformal methods quantify uncertainty estimates on top of the point predictions made by the mean estimator, altering the mean estimator will cause fluctuations in performance. To illustrate this fact, we used a more recent backbone model, ED-HNN Wang et al. (2023a), that had slightly lower validation accuracy than HCHA **Congress** and slightly higher validation accuracy on **House-Bills** datasets. The results of our experiment are shown in Table 8.

The experimental results show that using ED-HNN instead of HCHA produces wider uncertainty estimates for the **Congress** dataset. On the other hand, for the **House-Bills** dataset, we observe slightly shorter predictive bands. This empirically validates the correlation between the predictive performance of the mean estimator and the size of the uncertainty bands. The higher the predictive accuracy of the mean estimator, the shorter the size of the uncertainty bands, and vice versa.

## A.10 SCALABILITY OF CF-HGNN AND OUR METHOD

The limitation of both our method and CF-GNN is that they are more computationally expensive. However, this comes at a cost of shorter predictive bands (as the losses encourage shorter band length while maintaining desired levels of coverage). On that front, both CF-GNN and our method do not take too much time for this optimization in the calibration step. To demonstrate this, we perform an

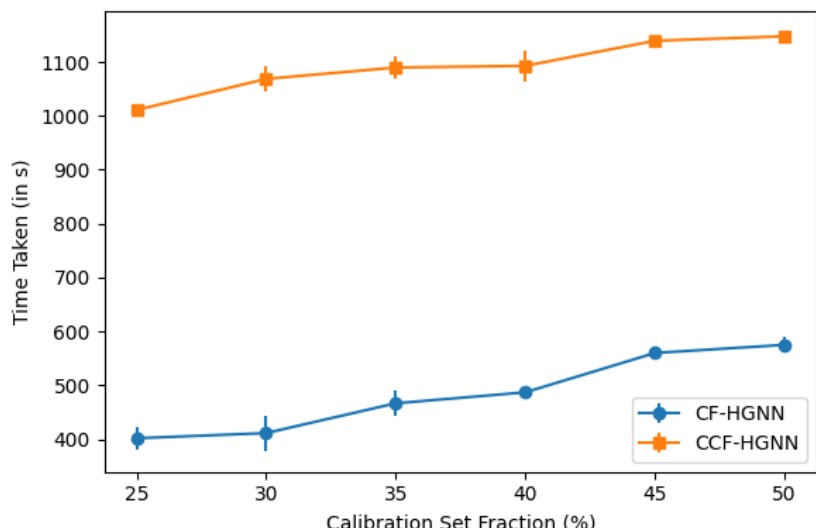

Figure 5: Comparison of total time taken for calibration optimization for both models on different calibration set sizes (averaged across 20 runs).

additional experiment on **Walmart-Trips** dataset by altering the size of the calibration set and noting the total time taken for calibration for 5000 epochs(in s). The results are shown in Figure 5.

While our method takes 2.5x more time than CF-HGNN (due to additions), our method scales by the same rate when increasing the calibration set size. This cost is offset by the improved performance in producing the uncertainty bands.

## A.11 EXAMPLES OF VIOLATION OF ASSUMPTION 1 IN THEOREM 2

There are some extreme examples where the contrastive augmentations (structure-altering augmentations) will violate this assumption. They are, as follows:

1. In multi-class node classification with extreme label imbalance, where even small augmentation may disproportionately isolate nodes from minority classes. In such cases, the topology-aware correction mechanism may no longer propagate reliable information through the local neighborhood, causing calibration to break down and resulting in lower empirical marginal coverage on average.

2. Another example occurs in hypergraphs with extremely poor connectivity, such as containing a single bridging hyperedge that connects two or more large, otherwise disconnected hypergraph components. If a contrastive augmentation removes or perturbs this bridging hyperedge, the connectivity between the components is disrupted. As a result, the local neighborhood information used in the topology-aware correction may no longer reflect the true label dependencies across the hypergraph, potentially violating the Bounded Coverage assumption and leading to miscalibrated prediction sets.

