# OpenReview forum: "Conformalized Predictions in Hypergraph Neural Networks via Contrastive Learning"
_ICLR.cc/2026/Conference — Submitted to ICLR 2026_

### Official Review · Reviewer_GCfZ · 2025-10-24

**Soundness:** 3
**Presentation:** 3
**Contribution:** 3
**Rating:** 8
**Confidence:** 4

**Summary:**

This paper introduces two methods for conformal prediction on hypergraph neural networks (HGNNs): CCF-HGNN and CF-HGNN. As noted by the authors, CF-HGNN is a naive extension of CF-GNN (Huang et al., 2024) for hypergraphs, which they include as a baseline for CCF-HGNN. CCF-HGNN employs two additional mechanisms—contrastive augmentations and a loss and degree prediction head—to account for aleatoric uncertainty.

While CF-HGNN is not part of the authors' main contribution, I believe it is still a valuable contribution, as it gives another CP method for HGNNs that achieves coverage requirements and is likely more scalable (see Weaknesses).

**Strengths:**

- This paper deals with an important and relevant area, which is uncertainty quantification for HGNNs
- The paper accounts for **both** aleatoric and epistemic uncertainty, which is missing in prior (graph) CP literature
- Both methods achieve the desired coverage guarantees and outperform baselines

**Weaknesses:**

- One limitation of CF-GNN compared to other graph CP methods (e.g., DAPS, NAPS), it is a lot more computationally expensive (as you are training a separate conformal model). Could you discuss how CF-HGNN and CCF-HGNN scale
- Two of the four datasets used were for binary classification. While CP still works, it is less meaningful when the only set options are nothing, either a 0/1 label, or everything. It would be nice to have seen more multi-class datasets. There seem to be several datasets here (under Hypergraphs with labeled nodes): https://www.cs.cornell.edu/~arb/data/

**Questions:**

- In Assumption 1, do you mean to say permute calibration and test *nodes*, rather than *edges*? Should L165 end with be (..., $\mathcal{V} _ {\pi}$,
$\mathcal{E} _ {\pi}$) rather than (...,
$\mathcal{V}$,
$\mathcal{V} _ {\pi}$)
- For your Bounded Coverage assumption for Theorem 1, do you have any examples of contrastive augmentations that would violate this assumption? It would be nice to include those in the manuscript
----
- Nit: Algorithm 1 has some abuse of notation ($\alpha$ is attention and significance level), which makes it a bit confusing

---

> ### Author Response · Authors · 2025-11-19
> **Rebuttal by Authors**
>
> We appreciate the reviewer's insightful comments and are encouraged by the positive feedback and recognition of our work's significance. We have included the results of all the additional experiments in the manuscript and hope that they improve the quality of our submission. We answer the questions and provide a response to the weaknesses below:
>
> ---
>
> ### **How CF-HGNN and CCF-HGNN scale**
>
> You are correct in saying that the limitation of both our method and CF-GNN is that they are more computationally expensive. However, this comes at a cost of shorter predictive bands (as the losses encourage shorter band length while maintaining desired levels of coverage). On that front, both CF-GNN and our method do not take too much time for this optimization in the calibration step. To demonstrate this, we perform an additional experiment on **Walmart-Trips** dataset by altering the size of the calibration set and noting the total time taken for calibration for 5000 epochs(in s). The results are as follows:
>
> | **Fraction** | **CF-HGNN**        | **CCF-HGNN**         |
> |---|---|---|
> | 25% | 401.72 ± 20.46 | 1010.72 ± 11.98 |
> | 30% | 411.25 ± 33.00 | 1068.51 ± 24.20 |
> | 35% | 466.57 ± 23.80 | 1089.59 ± 21.30 |
> | 40% | 486.91 ± 10.70 | 1092.70 ± 28.80 |
> | 45% | 559.91 ± 9.80  | 1139.25 ± 11.35 |
> | 50% | 574.99 ± 14.49 | 1147.57 ± 7.79  |
>
> While our method takes 2.5x more time than CF-HGNN (due to additions), our method scales by the same rate when increasing the calibration set size.
>
> ---
>
> ### **Nice to have seen more multi-class datasets**
>
> We have used 2 additional datasets from the link (contact-high-school and trivago-clicks) and evaluated our methods. Note that for trivago clicks, the number of unique classes was 160, with many singleton classes. In our experiment, we included the nodes that were a part of the top 80 of the label counts. The absence of the number of nodes in contact-high-school means that the uncertainty bands are large. The results are as follows:
>
> | Dataset | Model | APS Coverage | APS Length | RAPS Coverage | RAPS Length |
> |---|---|---|---|---|---|
> | High-School | CP | 97.86 ± 0.45 | 7.81 ± 0.07 | 96.14 ± 0.00 | 7.66 ± 0.04 |
> | High-School | CF-HGNN | 97.33 ± 0.01 | 7.44 ± 0.06 | 95.98 ± 0.06 | 7.28 ± 0.05 |
> | High-School | Ours | 97.79 ± 0.00 | **7.30 ± 0.32** | 96.13 ± 0.00 | **7.13 ± 0.73** |
> | Trivago-clicks | CP | 95.08 ± 0.04 | 56.23 ± 2.55 | 95.11 ± 0.04 | 54.79 ± 1.41 |
> | Trivago-clicks | CF-HGNN | 95.03 ± 0.00 | 52.09 ± 1.04 | 95.01 ± 0.03 | 51.38 ± 2.01 |
> | Trivago-clicks | Ours | 95.11 ± 0.04 | **51.47 ± 1.44** | 95.11 ± 0.00 | **50.89 ± 1.66** |
>
> As you can see, our method outperforms the closest baseline in both datasets.
>
> ---
>
> ### **In Assumption 1, do you mean to say permute calibration and test nodes, rather than edges?**
>
> Yes, you are correct, nice catch! This was a typo from our side. We have fixed this.
>
> ---
>
> ### **For your Bounded Coverage assumption for Theorem 1, do you have any examples of contrastive augmentations that would violate this assumption?**
>
> There are some extreme examples where the contrastive augmentations (structure-altering augmentations) will violate this assumption. They are, as follows:
>
> - In multi-class node classification with extreme label imbalance, where even small augmentation may disproportionately isolate nodes from minority classes. In such cases, the topology-aware correction mechanism may no longer propagate reliable information through the local neighborhood, causing calibration to break down and resulting in lower empirical marginal coverage on average.
>
> - Another example occurs in hypergraphs with extremely poor connectivity, such as containing a single bridging hyperedge that connects two or more large, otherwise disconnected hypergraph components. If a contrastive augmentation removes or perturbs this bridging hyperedge, the connectivity between the components is disrupted. As a result, the local neighborhood information used in the topology-aware correction may no longer reflect the true label dependencies across the hypergraph, potentially violating the Bounded Coverage assumption and leading to miscalibrated prediction sets.
>
> ---
>
> ### **Algorithm 1 has some abuse of notation**:
>
> Thanks for pointing this out. We have fixed this to avoid confusion.
>
> ---
>
> Thank you for posing these important questions. The results of the experiments have been added to the manuscript. Please let us know if there are more clarifications that we can address to improve the quality of our manuscript.

---

> > ### Comment · Reviewer_GCfZ · 2025-11-19
> >
> > Thank you for addressing my questions and concerns. I believe your paper can add value to the conference, so I will maintain my score. I look forward to reading an updated camera-ready version!

---

> > > ### Author Response · Authors · 2025-11-20
> > >
> > > Thank you for your comments and positive score. We have updated the manuscript. Please let us know if you have any concerns.

---

### Official Review · Reviewer_WUvR · 2025-10-27

**Soundness:** 2
**Presentation:** 3
**Contribution:** 2
**Rating:** 4
**Confidence:** 4

**Summary:**

This paper works on uncertainty quantification of hypergraph neural networks. The paper proposes a Contrastive Conformal
HGNN (CCF-HGNN) that jointly accounts for aleatoric and epistemic uncertainties in hypergraph-based models for guaranteed and robust uncertainty estimates. Compared with previous methods, it accounts for aleatoric uncertainty by leveraging contrastive learning on the structure of the hypergraph. The authors provide extensive experiments to show the effectiveness of the method.

**Strengths:**

1. The challenges of uncertainty quantification on hypergraph neural networks are clearly shown and explained. And the proposed method appropriately addresses all the listed challenges.
2. Theoretical and quantitative analyses are provided to show the rigorousness of the work.
3. The paper is well written and organized.

**Weaknesses:**

1. This paper argues that aleatoric uncertainty is not covered in the conformalized methods. This confuses me, as based on my understanding, the epistemic and aleatoric uncertainty have already been covered in the conformal prediction. Please explain why the CF-GNN or other conformal prediction methods do not cover aleatoric uncertainty.

2. The baselines in this paper are not enough. The authors argue that structural high-order information is considered to address the challenges in uncertainty quantification of hypergraph neural networks. Then, methods such as [1] should also be included as a competitive baseline that have already considered high-order structural information.

3. What is the backbone model used in the experiments? I understand that this paper focuses on the uncertainty of hypergraph neural networks. But will different hypergraph models have different influences on the results?

[1] Soroush H Zargarbashi, Simone Antonelli, and Aleksandar Bojchevski. 2023. Conformal prediction sets for graph neural networks. In the International Conference on Machine Learning. PMLR, 12292–12318.

**Questions:**

Please refer to the weaknesses.

---

> ### Author Response · Authors · 2025-11-19
> **Rebuttal by Authors**
>
> We deeply appreciate your valuable comments and recognize that the challenges of UQ in hypergraph neural networks are substantial. We will incorporate these suggestions into the final version. We are certain they will substantially improve the presentation of our work. Please find our response to your questions below:
>
> ---
>
> ### **Why the CF-GNN or other conformal prediction methods do not cover aleatoric uncertainty**
>
> While CF-GNN or other conformal predictions quantify uncertainty as a whole, they do not specifically distinguish between aleatoric and epistemic uncertainty. CF-GNN focuses on improving APS non-conformity scores, which effectively sharpens predictions by reducing epistemic uncertainty. As the calibration size increases, this approach can lead to sharper predictions, consequently reducing the overall inefficiency while maintaining coverage, but not explicitly accounting for aleatoric uncertainty.
>
> On the other hand, in our work, the contribution of aleatoric uncertainty due to structural noise is explicitly controlled by contrastive learning. By altering the augmentation rate, we can effectively control the rate of structural aleatoric uncertainty and thus can learn to minimize predictive band length while maintaining coverage.
>
> ---
>
> ### **The baselines in this paper are not enough**
>
> Please note the paper you have noted, DAPS [1] and similar work like SNAPS [2] propose a non-conformity score that can better represent the structure of the graphs. Our work (and approaches like CF-GNN [3]) is orthogonal to these works, as we focus on boosting any given non-conformity score with network information. Hence, DAPS can serve as an alternative for APS in our experiments.
>
> Based on your suggestion, we conducted the experiments on the **Walmart-Trips** and **DBLP** datasets, and the results are in the table below. We will include this result in the appendix.
>
> ---
>
> ### **Results (Walmart-Trips and DBLP)**
>
> | Dataset        | Method          | Coverage             | Length              |
> |---------------|-----------------|----------------------|---------------------|
> | Walmart-Trips | CP-APS          | 95.16 ± 0.02         | 8.973 ± 0.003       |
> | Walmart-Trips | CP-RAPS         | 95.08 ± 0.01         | 8.901 ± 0.053       |
> | Walmart-Trips | CF-HGNN-APS     | 95.05 ± 0.02         | 8.588 ± 0.003       |
> | Walmart-Trips | CF-HGNN-RAPS    | 95.01 ± 0.00         | 8.611 ± 0.001       |
> | Walmart-Trips | Our-APS         | 95.07 ± 0.03         | **8.518 ± 0.001**       |
> | Walmart-Trips | Our-RAPS        | 95.05 ± 0.01         | 8.592 ± 0.003       |
> | DBLP          | CP-APS          | 97.05 ± 0.00         | 3.470 ± 0.065       |
> | DBLP          | CP-RAPS         | 95.07 ± 0.00         | 1.600 ± 0.011       |
> | DBLP          | CF-HGNN-APS     | 97.11 ± 0.01         | 3.920 ± 0.011       |
> | DBLP          | CF-HGNN-RAPS    | 95.08 ± 0.00         | 1.703 ± 0.002       |
> | DBLP          | Our-APS         | 98.79 ± 0.01         | 3.734 ± 0.067       |
> | DBLP          | Our-RAPS        | 95.13 ± 0.00         | **1.508 ± 0.067**       |
>
> We notice that for Walmart-Trips, using DAPS does not improve performance for our method. However, the performance of our method improves when evaluated on the DBLP dataset. This is primarily because the non-conformity score of DAPS induces homophily and thus does not improve performance in a heterophilous hypergraph like Walmart-Trips. For a homophilous hypergraph like DBLP, performance improves significantly for all methods when using a more appropriate non-conformity score.
>
> ---
>
> ### **What is the backbone model used in the experiments?**
>
> The backbone model used in our experiments is **HCHA** [4].
>
> ---
>
> [1] Zargarbashi, Soroush H., Simone Antonelli, and Aleksandar Bojchevski. "Conformal prediction sets for graph neural networks." International Conference on Machine Learning. PMLR, 2023.
>
> [2] Song, Jianqing, et al. "Similarity-navigated conformal prediction for graph neural networks." Advances in Neural Information Processing Systems 37 (2024): 48541-48567.
>
> [3] Huang, Kexin, et al. "Uncertainty quantification over graph with conformalized graph neural networks." Advances in Neural Information Processing Systems 36 (2023): 26699-26721.
>
> [4] Bai, Song, Feihu Zhang, and Philip HS Torr. "Hypergraph convolution and hypergraph attention." Pattern Recognition 110 (2021): 107637.

---

> > ### Author Response · Authors · 2025-11-19
> > **Rebuttal Cont.**
> >
> > ### **Will different hypergraph models have different influences on the results?**
> >
> > Yes, different hypergraph models will have different influences on the results, as the method wholly relies on the non-conformity scores, which rely on the point predictions made by the model. However, this behavior will be observed in all methods.
> >
> > To illustrate this, we also used a more recent backbone model, **ED-HNN** [5], which had slightly lower validation accuracy than HCHA, on two datasets. The results are shown below.
> >
> > ---
> >
> > ### **Results with ED-HNN Backbone**
> >
> > | Dataset      | Model | APS Coverage      | APS Length       | RAPS Coverage     | RAPS Length      |
> > |--------------|-------|------------------|------------------|-------------------|------------------|
> > | Congress     | CP    | 99.49 ± 0.00     | 1.95 ± 0.00      | 95.29 ± 0.00      | 1.78 ± 0.12      |
> > | Congress     | Ours  | 98.99 ± 0.00     | 1.97 ± 0.02      | 95.28 ± 0.00      | 1.40 ± 0.05      |
> > | House-Bills  | CP    | 99.53 ± 0.00     | 1.95 ± 0.01      | 95.25 ± 0.00      | 1.24 ± 0.07      |
> > | House-Bills  | Ours  | 98.74 ± 0.01     | 1.96 ± 0.03      | 95.19 ± 0.00      | **1.15 ± 0.15**      |
> >
> > ---
> >
> > We hope that our responses clarify your concerns. Please let us know if you have further questions that we can clarify.
> >
> > ---
> >
> > [5] Wang, Peihao, et al. "Equivariant Hypergraph Diffusion Neural Operators." The Eleventh International Conference on Learning Representations.

---

> > > ### Author Response · Authors · 2025-11-24
> > > **Follow Up**
> > >
> > > Hi Reviewer WUvR,
> > >
> > > Please let us know if there are additional concerns we can address.
> > >
> > > Regards,
> > >
> > > Authors of Submission 21680.

---

> ### Comment · Reviewer_WUvR · 2025-11-25
>
> Thanks for the rebuttal. The answers address my concern. I'd like to maintain my score.

---

### Official Review · Reviewer_jvNm · 2025-10-31

**Soundness:** 2
**Presentation:** 3
**Contribution:** 2
**Rating:** 4
**Confidence:** 3

**Summary:**

This paper tackles how to give reliable uncertainty for hypergraph neural networks. The authors propose CCF-HGNN, which combines two ideas: (1) conformal prediction (APS/RAPS) to guarantee that the true label is covered, plus a topology-aware, differentiable training trick to make the prediction sets as small as possible; and (2) contrastive learning with simple structural augmentations (dropping hyperedges or edges) so the model learns representations that are robust to noisy structure (handling data uncertainty). They also add a lightweight auxiliary task that predicts hyperedge degrees, using attention and differentiable top‑k selection to focus on the most informative hyperedges, which further sharpens the predictions.

**Strengths:**

1. The problem is important and timely: uncertainty quantification for HGNNs is underexplored yet crucial for reliable deployment.

2. The proposed method appears reliable, with sound design choices (conformal calibration, contrastive robustness, auxiliary structural task) and both theoretical and empirical support.

3. The paper clearly attributes techniques to prior work, allowing readers to trace components (e.g., conformal prediction, contrastive learning, attention, Gumbel-Softmax) to their sources.

**Weaknesses:**

The method is largely a composition of existing techniques, as the authors themselves acknowledge (combining aleatoric via contrastive augmentation and epistemic via conformal prediction). The conformal setup closely follows Huang et al. (2024), the contrastive augmentations draw from Wei et al. (2022), and the auxiliary degree prediction leverages standard attention and differentiable top-k sampling. While the integration is well-executed, the incremental novelty—particularly relative to Huang et al. (2024) on the conformal side—feels modest. I am less familiar with hypergraph-specific precedents, but within conformal prediction the step beyond prior art seems mild.

**Questions:**

In Table 2, APS coverage appears substantially above the target 0.95 across multiple datasets. While APS can be conservative in practice, the magnitude and consistency of the overshoot suggest a potential calibration or implementation issue.

---

> ### Author Response · Authors · 2025-11-19
> **Rebuttal by Authors**
>
> We thank the reviewer for the constructive feedback and for recognizing the need for UQ methods for reliable hypergraph neural network deployment. We appreciate the thoughtful questions and critiques posed and address them below:
>
> ---
>
> - **The incremental novelty—particularly relative to Huang et al. (2024) on the conformal side—feels modest**:
>
>     As you have pointed out, our work takes inspiration from CF-GNN [1] (which is quite standard in hypergraph neural network literature; for example, HGNN [2] generalizes the message-passing scheme in GNNs [3], while HCHA [4] builds on top of GAT [5]). While we have readily attributed prior literature for better readability, we believe that our contributions are significant due to the following aspects:
>
>     1. CF-GNN exploits the idea that the local topology in graphs can influence the uncertainty estimates of a node.
>        However, the only way to extend this idea to a hypergraph is to induce message passing through all nodes connected via a hyperedge (which is essentially assuming the nodes within a hyperedge form a clique). Note that simply propagating uncertainties through the cliques will lead to mostly similar uncertainties for the nodes connected by a hyperedge, which is restrictive.
>
>     2. To solve this problem, perturbing the structure of the hypergraph and utilizing the InfoNCE loss for self-supervised learning encourages diversity in the uncertainty estimates, which improves the overall band length (as shown by our results).
>
>     3. Additionally, this approach leads to a theoretical guarantee on the prediction set size. Lemma 2 shows that the mutual information, $I(Y;\mathbf{Z}_1)$, between the label and the latent embedding learned by our method is guaranteed to be higher than the mutual information, $I(Y;\mathbf{Z}_0)$, between the label and the latent embedding learned by a simple extension of CF-GNN to hypergraphs. We believe that this integration is more principled and novel.
>
> ---
>
> - **APS coverage appears substantially above the target of 0.95:**
>
>     We carefully verified our APS implementation. It includes the required randomized smoothing term $U \sim \mathrm{Unif}(0,1)$ and uses the exact conformal quantile $\lceil (1-\alpha)(n_{\text{cal}}+1) \rceil$ applied to the multiset of calibration scores together.
>
>     Please note that while the coverages for APS are $0.99$ for some of our datasets, including DBLP, Congress, and House-Bills, it is closer to $0.95$ for Walmart-Trips, High-School, and Trivago-clicks datasets **(see our response to reviewer GCfZ)**. In the first set of datasets, two factors amplify the conservativeness. First, the House-Bills and Congress dataset exhibits highly concentrated class-probability distributions, causing APS to include multiple labels before the calibration threshold is reached. Second, the calibration fold size in the transductive split induces a slight upward bias in the empirical quantile, which further increases coverage. To address this issue, we also report results using RAPS, which adds a regularization term that penalizes unnecessarily large prediction sets. As shown in the experiments, RAPS yields coverage much closer to the target while substantially improving efficiency.
>
> ---
>
> We hope that our responses clarify your concerns. We have added additional experimental results in the revised manuscript. Please let us know if you have additional questions, clarifying which may improve the quality of this work.
>
> [1] Huang, Kexin, et al. "Uncertainty quantification over graph with conformalized graph neural networks." Advances in Neural Information Processing Systems 36 (2023): 26699-26721.
>
> [2] Feng, Yifan, et al. "Hypergraph neural networks." Proceedings of the AAAI conference on artificial intelligence. Vol. 33. No. 01. 2019.
>
> [3] Kipf, T. N., and M. Welling. "Semi-supervised classification with graph convolutional networks, in the Int." Conf. on Learning Representations. 2016.
>
> [4] Bai, Song, Feihu Zhang, and Philip HS Torr. "Hypergraph convolution and hypergraph attention." Pattern Recognition 110 (2021): 107637.
>
> [5] Veličković, Petar, et al. "Graph attention networks." arXiv preprint arXiv:1710.10903 (2017).

---

> > ### Author Response · Authors · 2025-11-24
> > **Follow Up**
> >
> > Hi Reviewer jvNm,
> >
> > Please let us know if there are additional concerns we can address.
> >
> > Regards,
> >
> > Authors of Submission 21680.

---

> ### Comment · Reviewer_jvNm · 2025-11-26
>
> Thanks for authors' rebuttal. I am still concern about novelty and the result of APS. I decide to maintain the score.

---

### Author Response · Authors · 2025-11-19
**General Responses to All Reviewers**

We would like to thank the reviewers for taking the time to read our paper and pose valuable questions and critiques. We have tried to address the questions asked in our revised version. The summary of our revisions is as follows:

- Fixed Notation /text ambiguities (Assumption 1 and Algorithm 1).

- Moved Sensitivity Study 3 from the Appendix to the Main Text

- Added the performance of Multi-Class Hypergraph Datasets and added descriptions of 2 new datasets in Appendix A.5.

- Added results of DAPS in the Appendix A.8.

- Revised the Proof of Lemma 2 in Appendix A.2 for clarity.

- Added results of ED-HNN in Appendix A.9.

- Added scalability results in Appendix A. 10.

- Added examples of violation of Assumption 1 in Theorem 2 in Appendix A.11.

---

We hope that this addressed the concerns of all the reviewers. Please let us know if we can improve the quality of the manuscript further and let us know if you have further concerns we can address.

Regards,

Authors of Submission 21680.

---

### Author Response · Authors · 2025-12-03
**Summary of the Rebuttal**

Dear ICLR 2026 AC, SAC, and PC,

We express our sincere gratitude to all reviewers for their valuable feedback and comprehensive initial reviews. We appreciate the reviewers' recognition of the strengths of our work, including clear explanations regarding the challenges in uncertainty quantification for hypergraphs (**Reviewers GCfZ, WUvR and jvNm**), sound design choices with theoretical and empirical support (**Reviewers WUvR and jvNm**), and overall clarity and organization of the manuscript (**Reviewers GCfZ, and WUvR**).

One major concern was the scalability of both CF-HGNN and CCF-HGNN (**Reviewer GCfZ**). We addressed this by adding an empirical analysis of the overall runtime for both CF-HGNN and CCF-HGNN by altering the size of the calibration set (from line 1183 to line 1215 in the revised paper). These results demonstrate that while CCF-HGNN takes 2.5x more time than CF-HGNN (due to additional losses), it scales by the same rate when increasing the calibration set size compared to CF-HGNN. Based on the suggestion of Reviewer GCfZ, we also performed experiments on two additional multi-class hypergraph datasets (High-School and Trivago-Clicks) that demonstrated the utility of our proposed method in multi-class datasets (from line 473 to line 493 in the revised paper). Another concern was the comparison with existing non-conformity scores like DAPS (**Reviewer WUvR**). We clarified that our work and CF-GNN are orthogonal to prior works on graphs like DAPS and SNAPS as we focus on boosting any given non-conformity score with network information instead of defining a non-conformity score for networks (like DAPS and SNAPS). We added results for all the conformal methods by optimizing DAPS for Walmart-Trips and DBLP datasets (from line 1125 to line 1158 in the revised paper), which showed that for Walmart-Trips, using DAPS does not improve performance for our method. However, the performance of our method improves when evaluated on the DBLP dataset. Additionally, we performed a sensitivity study (from line 1159 to line 1182 in the revised paper) by altering the mean estimator from HCHA to ED-HNN that showed that the choice of the mean estimator effects the size of the predicted uncertainty bands of all methods.

We have added experiments and analysis in the revised version according to the reviewers' suggestions and concerns to improve accessibility for a broader audience. Additionally, we made the following changes:

- **Fixed Notation text ambiguities (Assumption 1 and Algorithm 1).**
- **Moved Sensitivity Study 3 from the Appendix to the Main Text.**
- **Revised the Proof of Lemma 2 in Appendix A.2 for clarity.**
- **Added examples of violation of Assumption 1 in Theorem 2 in Appendix A.11.**

We respect the concern on the incremental novelty, particularly relative to Huang et al. (2024) on the conformal side (**Reviewer jvNm**). However, as we emphasized during the rebuttal, our novelty lies in the principled integration of hypergraph-specific structural perturbations with a contrastive learning objective to overcome the limitations of directly extending CF-GNN to hypergraphs. By encouraging diversity in latent representations and uncertainty estimates—rather than collapsing uncertainties within each hyperedge—we achieve both improved empirical band lengths and a theoretical guarantee showing higher mutual information between labels and learned embeddings compared to the CF-GNN extension. This combination, to our knowledge, has not been explored in prior conformal or hypergraph learning work. Reviewer GCfZ further acknowledged that our paper can add value to the conference and they will look forward to reading the camera-ready copy of our manuscript.

Finally, we are grateful that reviewers highlighted the quality and potential impact of our work. We believe our contributions advance uncertainty quantification for hypergraphs, bridging theoretical insight with practical utility.

Regards,
The Authors of Submission 21680

---

### Meta-Review · Area_Chair_Mrv8 · 2026-01-12

**Summary:**

The paper introduces CCF-HGNN that uses conformal prediction to imbue hypergraph neural networks with uncertainty quantification capabilities. The authors also propose a topology-aware training to reduce set size, a simple contrastive learning loss, and an auxiliary task that predicts hyperedge degrees. The reviewers agree that the problem addressed is important and that the paper is well-written and easy to follow, with clear attribution to prior work.

Reviewer jvNm raises a concern about the novelty of the proposed method. In their rebuttal, the authors clarify how they went beyond prior work, and specifically highlight their theoretical contributions. While the observations made in Lemma 2 and Theorem 1 are certainly interesting, these are not technically novel or particularly deep results. I would even go so far as to say that these may be better characterised as propositions rather than lemmas/theorems. Reviewer jvNm also raises a concern about the implementation of APS since the empirical coverage is significantly higher than the nominal level. The authors attempt to provide an explanation in the rebuttal, but this is not entirely convincing. The reviewer explicitly states that they were not convinced by the authors' response, and I agree with this assessment.

The discussion of aleatoric and epistemic uncertainty throughout the paper is quite imprecise. Many experts in uncertainty quantification would like disagree with the way these terms are used in the paper. CP, does not explicitly model or capture epistemic uncertainty, but rather provides finite-sample valid prediction sets under exchangeability assumptions. It reflects both types of uncertainty as far as they are present in the data data, but it does not separate or interpret them. Therefore, statements such as "To the best of our knowledge, this is the first work that combines aleatoric uncertainty (contrastive augmentation- aided learning) and epistemic uncertainty (conformal prediction)." are not accurate.  Reviewer WUvR also raises an issue about the discussion of aleatoric uncertainty. The authors response is again not entirely convincing.

Reviewer WUvR asked to include additional baselines, and the authors provided a table but it's not clear which rows correspond to DAPS which was the requested baseline (same for Table 7 where DAPS is mentioned in the caption but not in the table).

While focusing on the transductive node classification problem is a good start, it should be acknowledged as a limitation, since many real-world applications require inductive capabilities.

Reviewer GCfZ had a more positive view of the paper and their main concern about scalability was sufficiently addressed in the rebuttal.

Overall, based on the reviews and the authors' responses, I recommend this paper to be rejected.

**Reviewer Concerns:**

Reviewer jvNm: Concerns about novelty and the potential issues with the implementation were not sufficiently addressed.

Reviewer WUvR: The matter-of-fact questions (e.g. backbone) were addressed. The discussion of aleatoric uncertainty was not sufficiently addressed, it's unclear whether the additional baseline was addressed.

Reviewer GCfZ: Sufficiently addressed the main concerns.

**Reviewer Scores:**

All the reviewers already state that they would like to maintain their score.

---

### Decision · Program_Chairs · 2026-01-26

Reject